# DominoSearch: Find layer-wise fine-grained N:M sparse schemes from dense neural networks

**Wei Sun**[†][*]    **Aojun Zhou**[‡][*]    **Sander Stuijk**[†]    **Andrew Nelson**[†]    **Rob Wijnhoven**[¶]
**Hongsheng Li**[‡]    **Henk Corporaal**[†]
[†]Eindhoven University of Technology   [¶]ViNotion B.V., the Netherlands
[‡]CUHK-Sensetime Joint Lab, CUHK
w.sun@tue.nl   aojunzhou@gmail.com
rob.wijnhoven@vinotion.nl   hsli@ee.cuhk.edu.hk
{s.stuijk,a.nelson,h.corporaal}@tue.nl

## Abstract

Neural pruning is a widely-used compression technique for Deep Neural Networks (DNNs). Recent innovations in Hardware Architectures (e.g. Nvidia Ampere Sparse Tensor Core) and N:M fine-grained Sparse Neural Network algorithms (i.e. every M-weights contains N non-zero values) reveal a promising research line of neural pruning. However, the existing N:M algorithms only address the challenge of how to train N:M sparse neural networks in a uniform fashion (i.e. every layer has the same N:M sparsity) and suffer from a significant accuracy drop for high sparsity (i.e. when sparsity > 80%). To tackle this problem, we present a novel technique – ***DominoSearch*** to find mixed N:M sparsity schemes from pre-trained dense deep neural networks to achieve higher accuracy than the uniform-sparsity scheme with equivalent complexity constraints (e.g. model size or FLOPs). For instance, for the same model size with 2.1M parameters (87.5% sparsity), our layer-wise N:M sparse ResNet18 outperforms its uniform counterpart by 2.1% top-1 accuracy, on the large-scale ImageNet dataset. For the same computational complexity of 227M FLOPs, our layer-wise sparse ResNet18 outperforms the uniform one by 1.3% top-1 accuracy. Furthermore, our layer-wise fine-grained N:M sparse ResNet50 achieves 76.7% top-1 accuracy with 5.0M parameters. This is competitive to the results achieved by layer-wise unstructured sparsity that is believed to be the upper-bound of Neural Network pruning with respect to the accuracy-sparsity trade-off. We believe that our work can build a strong baseline for further sparse DNN research and encourage future hardware-algorithm co-design work. Our code and models are publicly available at https://github.com/NM-sparsity/DominoSearch.

## 1   Introduction

Modern deep neural networks (DNNs) achieve remarkable success in various tasks such as computer vision and language processing at a cost of memory footprint, computation, energy and storage. These have all become bottlenecks of deploying DNNs on commodity hardware for real-world applications. To address these challenges, many model compression techniques have been proposed to optimize DNN models. Neural network pruning is one promising technique orthogonal to quantization [1–3], network architecture search [4–6] and knowledge distillation [7–9].

Recent innovations in hardware architectures (e.g. Nvidia Ampere Sparse Tensor Core [10]) and N:M fine-grained sparse DNN pruning algorithms [11, 12] reveal a promising research line of neural

---

[*]Equal contribution.

pruning that may achieve a high sparse ratio while maintaining a regular sparse structure. On the other hand, traditional unstructured sparsity can achieve a higher sparse ratio but cannot maintain a regular sparse structure, which makes it difficult for acceleration [13]. Traditional structured sparsity can achieve regular sparse structure but may not achieve high compression ratio[11]. Figure 1 illustrates traditional unstructured/structured pruning and N:M fine-grained structured pruning.

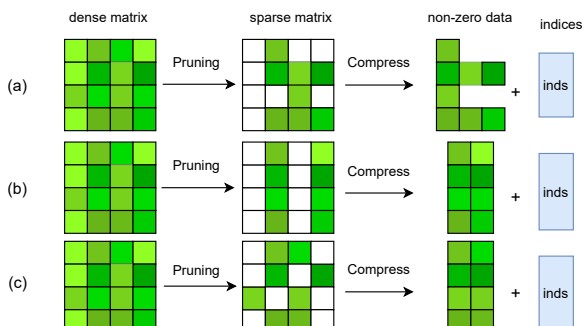

Figure 1: (a) Unstructured pruning treats every weight in the dense matrix individually which will produce a sparse matrix with irregular non-zero data. (b) Coarse-grained pruning treats a filter/channel as an atomic component and can produce a regular sparse matrix. (c) Fine-grained N:M pruning performs unstructured pruning inside each group (group size M = 4 in this example) while it enforces an additional constraint that there are at most N non-zero values to be kept.

Applying uniform fine-grained N:M sparsity training, or fine-tuning to DNNs, will lead to a sub-optimal solution because layers are treated equally. Modern DNNs tend to have a number of layers with a highly non-uniform distribution of redundancy. This well-known phenomenon has been exploited by recent unstructured pruning methods [14–17] that have achieved higher accuracy than the uniform counterparts, especially for relatively high sparsity. However, the existing methods mainly focus on unstructured pruning that treats the weights within a layer individually and does not have fine-grained position constraints, which cannot be used for fine-grained N:M sparsity. As a consequence, a novel method for layer-wise fine-grained N:M sparsity is needed.

Furthermore, sparsity acceleration relies on the underlying software-architecture system of the platform. We can make a reasonable assumption that there will be more but not arbitrary N:M schemes supported in future software-architecture designs. Although the flexibility of N and M is still an open system design research question, we assume a policy of fixed M and flexible N in this paper because it can provide fixed window size (i.e. size = M) for sparse weight matrix compression.

Motivated by above-mentioned challenges, this work introduces an efficient framework to find a layer-wise fine-grained N:M scheme with pre-defined N:M candidates which achieves higher accuracy compared to the uniform counterpart with equivalent complexity constraint (e.g. model size or FLOPs).

We propose DominoSearch, an iterative algorithm that finds layer-wise fine-grained sparse schemes from pre-trained dense weights. This is achieved using a magnitude-based criterion [14, 18] for the pruning selection threshold, in combination with a weight penalty to drive values towards the threshold. DominoSearch also uses a layer-wise penalty factor to balance the heuristic layer-wise redundancy of parameters[15] and layer-wise computational complexity (e.g. FLOPs). As indicated by the name *Domino*, the N value from the N:M ratio decrements during the search phase with the weight penalty acting as momentum.

Due to the layer-wise sparsity, our framework is able to achieve state-of-the-art (SOTA) accuracy-model size trade-offs when compared with unstructured sparsity and accuracy-FLOPs trade-offs when compared with structured sparsity. Figure 2 demonstrates the advantages of layer-wise N:M sparsity under both scenarios on ResNet50 [19]. Details can be found in Table 2.

The major contributions of our paper can be summarized as follows: **(1)** We present DominoSearch - A novel and easy-to-use framework to search the layer-wise sparse schemes from pre-trained dense weights with a specified model complexity constraint. Our method does not rely on expensive design space exploration such as Network Architecture Search (NAS) or Reinforcement Learning (RL).

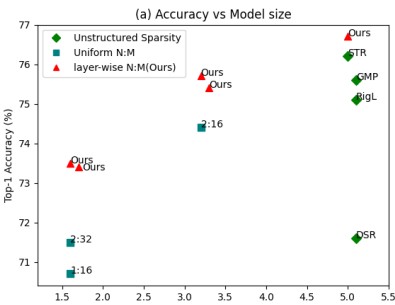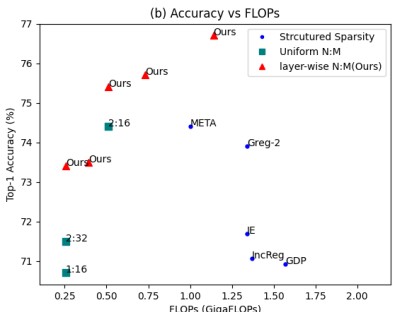

Figure 2: ResNet50 on ImageNet: Compare top-1 Accuracy of our layer-wise N:M sparse models with SOTA structured/unstructured sparsity under various FLOPs/model sizes.

To the best of our knowledge, this is the first work on layer-wise fine-grained N:M sparsity. **(2)** We propose a layer-wise penalty factor to balance the parameters redundancy that does not depend on input tensor/image size and complexity that depends on input tensor/image size (e.g. FLOPs). The computational complexity measurement FLOPs used in this paper can easily be replaced with latency, throughput or energy consumption when deployed on a specific platform. **(3)** We evaluate our solution on the large scale ImageNet dataset with both heavy ResNets [19] and compact RegNets [20], and the smaller scale CIFAR100 dataset with ResNet56. Results of ResNet50 on ImageNet show that layer-wise N:M sparsity searched by our framework achieves SOTA accuracy-compression trade-off under fine-grained N:M sparsity.

## 2 Related Work

**Structure of sparsity**: Traditionally, sparsity of DNNs is categorized into 1) unstructured sparsity and 2) structured sparsity. Unstructured sparsity can be achieved by weight pruning and reaches a relatively high sparse ratio without significant accuracy drop [14, 16, 18, 21, 22]. However, unstructured sparsity has a large execution time overhead on modern hardware due to the large overhead of handling the irregular non-zero weights in the matrix; good speed-up can only be achieved when the sparse ratio is very high (i.e., $> 95\%$) [13, 23]. On the other hand, coarse-grained structured sparsity such as filter-wise sparsity and channel-wise sparsity [23, 24] can maintain a hardware-friendly sparse structure but may not be able to achieve a high sparse ratio. In order to achieve a high compression ratio and a hardware-friendly sparse structure, various fine-grained structured sparsity patterns were proposed including Tile-wise [25], intragroup sparsity structure [26] and N:M sparsity [10]. Our paper focuses on N:M fine-grained sparsity which has demonstrated success in getting accurate N:M sparse DNNs (i.e. trained from scratch [11] or fine-tuned [27] from dense pretrained weights) and usage in commodity hardware (e.g. Nvidia Ampere Tensor Core). Figure 1 illustrates two traditional types (structured/unstructured) and fine-grained N:M sparsity.

**Layer-wise compression**: Layer-wise compression is a well-known concept in the DNN optimization community. The non-uniform importance of different layers in DNNs is leveraged by mixed-precision quantization in which different bit-widths are allocated via NAS [28, 29] or RL [3]. Recent discoveries demonstrate that by carefully selecting [14, 15] or learning [16] the layer-wise sparsity, the sparse DNNs can achieve higher accuracy than their uniform counterparts. Evci *et al.* [15] propose a training scheme for unstructured sparse neural networks and present the Erdos-Renyi kernel (ERK) (extension of [17]) to heuristically select the layer-wise sparsity. Lee *et al.* [14] propose a layer-adaptive magnitude-based pruning score to choose the layer-wise sparsity, focusing on unstructured pruning and the number of survival weights of each layer without taking the layer-wise computation complexity (e.g. FLOPs) into account. Kusupati *et al.* [16] present Soft Threshold Reparameterization (STR) to learn the layer-wise sparsity during training. However, the target model complexity constraint (e.g. model size and FLOPs) is implicitly controlled by two hyper-parameters ($\lambda$ and $s_{init}$) and cannot be specified, hence users have to tune $\lambda$ and $s_{init}$ and train multiple models to get the desired target model complexity. Our paper focuses on how to find the N:M scheme of each layer efficiently with the specified model complexity constraint. It avoids the need for a time consuming hyper-parameter search.

## 3 Method - DominoSearch

### 3.1 Problem Formulation

Given a pretrained dense model with weight tensors $\mathbf{w}^1, \ldots, \mathbf{w}^i, \ldots, \mathbf{w}^L$ associated with $L$ layers (convolution layers + fully-connected layers), model complexity constraint $Sp_t$ (e.g. model size or FLOPs) and candidate pool $N_p$ of sparse schemes (e.g. [1:8,2:8,4:8,8:8])[2]. The problem can be formulated as :

$$
\begin{aligned}
\max_{S(\mathbf{w},N)} \quad & \text{Accuracy}\big(S(\mathbf{w}, N)\big) \\
s.t. \quad & \text{Complexity}(S(\mathbf{w}, N)) \leq Sp_t \\
& N^i \in N_p
\end{aligned}
\tag{1}
$$

Where $S(\cdot)$ denotes the sparse model which can be obtained with dense weights $\mathbf{w}$ and sparse ratio $N$. Complexity$(\cdot)$ can be model size, FLOPs, real latency or even energy consumption depending on the use case and design objectives. The goal is to find optimal layer-wise schemes for the whole network. Due to the large discrete design space and the training/fine-tuning cost of evaluating a design point (i.e. recovering the accuracy of the sparse model), it is usually not possible to handcraft the optimal solution or to perform an exhaustive search.

Inspired by methods used in mixed-bit search [3, 29], which decouple mixed-bit policy searching and model accuracy recovering, we use a two-step solution: (1) find the layer-wise sparse schemes from the candidate pool $N_p$; (2) fine-tune/retrain sparse weights with searched sparsity (i.e. $N^i$) to recover the accuracy.

### 3.2 Group-wise threshold and sparsity

We first emphasise the difference between layer-wise fined-grained sparsity and layer-wise unstructured sparsity. How to decide the layer-wise sparsity has been actively studied over recent years [14–16]. However, these methods mainly focus on unstructured sparsity whose layer-wise sparsity can be decided by a layer-wise threshold because the weights in a certain layer are treated individually. In contrast, fine-grained N:M sparsity treats weights from a *group-wise* view (e.g. Figure 1 (c)). As a consequence, we define a group-wise threshold vector $T^i$ and group-wise sparsity vector $N_g^i$, where $i$ is the layer index. The length of these two vectors is $K = \frac{|\mathbf{w}^i|}{M}$, where $||$ is cardinality (i.e. the number of parameters), $M$ is the group size and $K$ is the number of groups. $|\mathbf{w}^i|$ should be divisible by $M$.

**Group-wise threshold selection**: Instead of using a small global value such as $\frac{1}{1000}$ [22] to decide if the weights should be treated as pruned or not, we use a threshold vector $T^i$ that can represent the weight distributions of each group. In order to achieve this, we use the mean of the smallest (absolute value) $\frac{M}{2}$ elements as threshold because it provides a group-wise feature in an efficient way and can mitigate the effect of outliers (e.g. values like 0.0 ). Note that we use $\frac{M}{2}$ elements because it results in a sufficient search space and representativeness. Using more elements can result in a better representativeness because more elements are included, but results in a smaller search space because a larger threshold value will lead to a smaller starting value $N_g^i$ and vice versa. Figure 3 shows an example of group-wise threshold selection. The larger variance of $T^i$ indicates that we need a group-wise threshold. On the other hand, a group-wise threshold will lead to group-wise sparsity. The column vector $N_g^i$ encodes the number of kept weights of each group, from which we can infer the sparsity of this layer (i.e. N:M = 3:4). In practice, we can relax the requirement by using the *Supermajority* voting rule. Specifically, as long as the ratio of the number of $x$ in $N_g^i$ reaches a pre-defined voting ratio $v_r$, the layer sparsity $N^i$ can be inferred as $x$.

With above-mentioned group-wise definitions, we can reformulate $\mathbf{w}^i$ and $T^i$ as $\mathbf{w}_k^i$ and $T_k^i$, where $k$ is the group index. We use the reformulated version in the next sections.

---

[2]Without losing generalization, we use N equal to power-of-two as search space.

$$
\overset{\mathbf{w}^i}{\begin{bmatrix} 0.0104 & 0.0114 & 0.0020 & 0.0061 \\ 0.0212 & 0.0748 & 0.0368 & 0.0898 \\ 0.0854 & 0.1751 & 0.0406 & 0.0450 \\ 0.0896 & 0.0169 & 0. & 0.0177 \end{bmatrix}} \;->\; \overset{T^i}{\begin{bmatrix} 0.0041 \\ 0.0290 \\ 0.0428 \\ 0.0085 \end{bmatrix}} \;->\; \overset{N_g^i}{\begin{bmatrix} 3 \\ 3 \\ 3 \\ 3 \end{bmatrix}}
$$

Figure 3: An illustrative example of dense weight matrix $\mathbf{w}^i$ and corresponding threshold vector $T^i$. The values are taken from random groups in **the first layer of ResNet18**. $N_g^i$ encodes the number of elements larger than the threshold of each group. Blue highlights values larger than threshold.

## 3.3 Explore the search space via layer-wise weight penalty

Before introducing our search algorithm, we consider the standard DNN training process with the Stochastic Gradient Descent (SGD) update rule:

$$\mathbf{w}(t+1) := \mathbf{w}(t) - \xi_t g(\mathbf{w}(t)) \tag{2}$$

with learning rate $\xi_t$ and gradients $g(\mathbf{w})$ of $\mathbf{w}$. To obtain a sparse model, we need to apply a mask $\mathbf{m} \in \{0,1\}$ to weights $\mathbf{w}(t)$, resulting in a sparse model $\widetilde{\mathbf{w}}(t) := \mathbf{w}(t) \odot \mathbf{m}$, where $\odot$ denotes element-wise product. The mask can dynamically change during training iteration $t$ and depends on the weights $\mathbf{w}_t$. With group-wise and layer-wise dimensions, we can reformulate the mask $\mathbf{m}$ as

$$\mathbf{m}_k^i(t) := \mathbf{w}_k^i(t) > T_k^i \tag{3}$$

where $i$ is the layer index, $k$ is the group index and $T_k^i$ is the group-wise threshold introduced in Section 3.2. Mask $\mathbf{m}_k^i(t)$ is inferred by Boolean Operator $>$, where $\mathbf{m} = 1$ means kept and $\mathbf{m} = 0$ means pruned.

A popular approach is to apply an extra regularization penalty to introduce more sparsity, including $L_1$ norm and $L_2$ norm [18, 22]. We periodically exploit the $L_2$ norm [22] which can hold the group-wise feature, to find the layer-wise sparsity. Thus the update rule can be formulated as:

$$\mathbf{w}_k^i(t+1) := \mathbf{w}_k^i(t) - \xi_t(g(\widetilde{\mathbf{w}}_k^i(t)) + p(t) * \eta_i(t) * \widetilde{\mathbf{w}}_k^i(t)) \tag{4}$$

$$p(t) = \begin{cases} 1, & \text{if } t \mod k_p = 0. \\ 0, & \text{otherwise.} \end{cases} \tag{5}$$

where $\eta_i(t)$ is the *layer-wise* penalty factor to control the strength of penalty applied to each layer which will be introduced in Section 3.4. Inspired by [22], we apply a **periodical** weight penalty which will drive the kept weights towards the threshold gradually and leave $k_p$ iterations as buffer time for adjusting to the applied penalty (using $p(t)$). The layer-wise sparsity $N^i$ and model complexity decrease during the search phase. When the model complexity reaches the search constraint $Sp_t$, we terminate the search process and fine-tune the weights based on the searched layer-wise fine-grained sparsity schemes. Algorithm 3.1 shows the search algorithm.

## 3.4 Layer-wise penalty factor

Modern DNNs tend to have many layers with non-uniform distributions of both parameter redundancy and complexity contribution. Intuitively, we want to encourage layers with larger redundancy and higher complexity to achieve a higher sparse ratio (i.e. decrease N) faster. This can be achieved by controlling the strength of the weight penalty of each layer. The layer-wise computational complexity (e.g. FLOPs) factor can be calculated using

$$c_i = \frac{f_i * s_i}{\max_{i=1}^{i=L}(f_i * s_i)} \tag{6}$$

where $f_i$ is the dense FLOPs of layer $i$ and $s_i = N_v^i/M$ is the sparsity (Line 10 in Algorithm 3.1).

**Algorithm 3.1** DominoSearch

---

1: **Input**: Complexity constraints $Sp_t$, dense weights of each layer $\mathbf{w}_k^1$, ..., $\mathbf{w}_k^l$. Dataset $D$.
   Learning rate $\xi$. Interval $K_c$. Voting ratio $v_r$.
2: **Init**: current model complexity $Sp_c = 0$, iterations $t = 0$, layer-wise penalty factor $\eta_i$.
3: **while** True **do**
4:     **for** $\mathbf{w}_k^i$ in $\mathbf{w}_k^1$, ..., $\mathbf{w}_k^L$ **do**
5:         $\mathbf{w}_k^i(t+1) := \mathbf{w}_k^i(t) - \xi_t(g(\widetilde{\mathbf{w}}_k^i(t)) + p(t) * \eta_i(t) * \widetilde{\mathbf{w}}_k^i(t))$          ▷ Eq. 4
6:         **if** t % $K_c$ == 0 **then**
7:             $\mathbf{m}_k^i(t) := \mathbf{w}_k^i(t) > T_k^i$          ▷ Eq. 3
8:             $N^i := $ vote( sum($\mathbf{m}_k^i(t)$, axis=1),$v_r$ )          ▷ Sec. 3.2
9:             **if** $N^i \in N_p^i$ **then**
10:                $N_v^i = N^i$          ▷ $N_v^i$ tracks $N^i \in N_p^i$
11:                $Sp_c = $ Complexity($N_v^i$)          ▷ Compute current model complexity
12:                **if** $Sp_c \geq Sp_t$ **then**
13:                    Terminate search process and go to **Output**.
14:                **end if**
15:                Update layer-wise penalty factor $\eta_i$          ▷ Sec. 3.4
16:             **end if**
17:         **end if**
18:     **end for**
19: **end while**
20: **Output** Layer wise sparse scheme $N_v^i$

---

In order to estimate the layer-wise redundancy, we use the Erdos-Renyi kernel (ERK) [15]. ERK is a heuristic method to decide the layer-wise unstructured sparsity given an overall sparsity target (i.e. model size) which does not require design space exploration or hyper-parameter search. However, ERK cannot be directly applied to select the layer-wise N:M schemes because: 1) ERK gives layer-wise sparsity in the continuous domain while N:M sparsity is discrete. 2) Layer-wise sparsity decided by ERK only depends on the overall sparsity target and layer configurations (e.g. kernel size, input/output channels). It can not reflect layer-wise complexity (e.g. FLOPs) which depends on input tensor/image size. Thus, we use the layer-wise sparsity generated by ERK as a layer-wise redundancy guidance, which can be formulated as:

$$r_i = \frac{e_i - s_i}{\max_{i=1}^{i=L} |e_i - s_i|} \tag{7}$$

where $e_i$ is the sparsity of layer $i$ decided by ERK. Note that $r_i$ can be negative if the current sparsity ($s_i$) is larger than the sparsity estimated by ERK. With $r_i$ and $c_i$, the joint *layer-wise* penalty factor $\eta_i$ is defined as:

$$\eta_i = \beta_1 * c_i + \beta_2 * r_i \tag{8}$$

where $\beta_1$ and $\beta_2$ are weighted coefficients ($\beta_1 + \beta_2 = 1$). When searching under model size constraint, we set $\beta_1 = 0.5$, $\beta_2 = 0.5$ and use $\beta_1 = 0.8$, $\beta_2 = 0.2$ for searching under FLOPs constraint. Weighted coefficients $\beta_1$ and $\beta_2$ are used to balance the effect of FLOPs which is dependent on input tensor/image size and model size which is not dependent on input tensor/image size, we tend to use higher $\beta_1$ when the complexity constraint is FLOPs because it can assign layers with higher FLOPs a higher penalty factor. An ablation study of *layer-wise* penalty factor can be found in Section 5.1.

## 4 Experimental Results

**Networks and datasets**: We demonstrate the advantages of layer-wise fine-grained N:M sparsity via various networks on the large-scale ImageNet dataset [30] including heavy ResNets [19] and compact RegNets [20]. We also demonstrate the effectiveness of our method on the smaller CIFAR100 dataset with ResNet56 [19].

**Training settings**: We first apply DominoSearch (DS) to find the layer-wise sparse schemes from pre-trained dense models (Algorithm 3.1), and fine-tune/retrain sparse weights on pre-trained dense weights with searched layer-wise schemes to recover the accuracy. Detailed settings can be found in the Supplementary Material.

## 4.1 ResNet18/50 with model size and FLOPs as constraints

**ResNet18**: Table 1 shows that layer-wise sparse ResNet18 consistently outperforms consistently outperforms the uniform baseline with both equivalent model size or FLOPs. Figure 4 visualizes the layer-wise sparse FLOPs distributions under two search targets and dense parameter distributions. For FLOPs-constrained search, the shallower layers, especially the first layer, are allocated with higher sparsity (Figure 4 (b)) because of their larger FLOPs contributions caused by the larger input feature size. On the other hand, shallower layers have relatively smaller contributions to the model size (i.e. less parameters, Figure 4 (c)), resulting in lower sparsity when using the model size constraint (Figure 4 (a)). This observation supports the necessity of using layer-wise penalty factor $\eta_i$ which is able to capture the information of FLOPs distribution from a global view and assign corresponding layers (shallower layers in this case) with relatively higher penalty factors.

Table 1: ResNet18. Layer-wise sparse vs. uniform baseline with equivalent model size or FLOPs. Dense baseline reference is from Pytorch pre-trained model zoo

| Method | #Params | FLOPs | Top1 Acc | Structure |
|---|---|---|---|---|
| ResNet18 | 11.7M | 1814M | 69.8 | Dense |
| SR-STE [11] | 1.46M | 227M | 66.65 | 2:16 |
| Equal model size | | | | |
| DS(**ours**) | 1.46M | 329M | 68.76(+2.1) | Mixed N:16 |
| Equal FLOPs | | | | |
| DS(**ours**) | 1.29M | 227M | 67.98(+1.3) | Mixed N:16 |

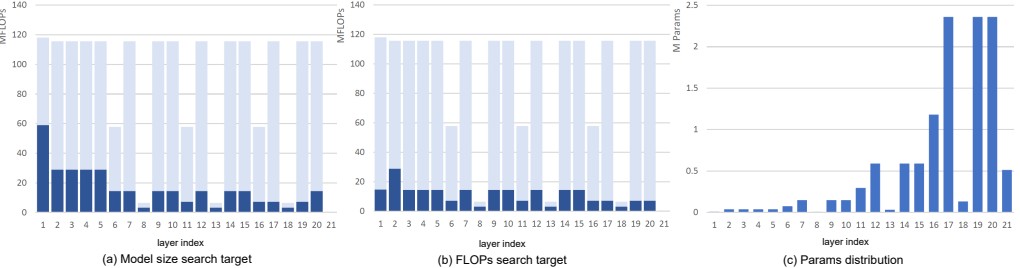

Figure 4: Layer-wise statistics of ResNet18: x-axis is layer index $i$ (layer 8, 13, 18 are *Conv2d* with $1 \times 1$ kernel and layer 21 is fully-connected layer) and y-axis denotes FLOPs and Parameters. (a)/(b) shows the distributions of layer-wise FLOPs under model size/FLOPs search target, where dark/light means kept/pruned partition. (c) shows the layer-wise distribution of parameters of dense ResNet18.

**ResNet50**: Table 2 shows the results for ResNet50. Layer-wise fine-grained sparsity can consistently outperform its uniform counterparts for both accuracy-sparsity (i.e. model size) and accuracy-FLOPs trade-off. When comparing to the SOTA layer-wise unstructured sparsity [16], which is believed to be the upper-bound of Neural Network pruning with respect of accuracy-sparsity trade-off, our layer-wise fine-grained models can achieve competitive result given the same model size. Although it has higher FLOPs than unstructured sparsity (1.14G v.s. 766M), we emphasize that fine-grained sparsity maintains a ***regular*** sparse pattern which is crucial for getting deployment gains (e.g. latency). Furthermore, our layer-wise fine-grained sparsity is significantly better than structured sparsity with respect to accuracy-FLOPs trade-off, which encourages a promising system design research line: efficient supports of flexible N:M sparse patterns. Considering the preliminary success of Nvidia Ampere Sparse Tensor Core [10], we believe the design overhead of supporting flexible N:M sparse patterns is affordable and worthwhile.

Table 2: ResNet50 performance. (+) denotes the top1-accuracy gain compared to the uniform counterparts with equivalent model size or FLOPs. (+) compares with unstructured sparsity. Reported numbers are either taken from the corresponding literature or obtained by training with open-source code [11]. Dense baseline reference is from Pytorch pre-trained model zoo.

| Method | #Params | FLOPs | Top1 Acc | Structure | Uniform |
|---|---|---|---|---|---|
| ResNet50 | 25.6M | 4.09G | 76.1 | Dense | - |
| GMP[31] | 5.1M | 818M | 75.6 | Unstructured | U(niform) |
| DSR[32] | 5.1M | 1.23G | 71.6 | Unstructured | L(ayer-wise) |
| DNW[33] | 5.1M | 818M | 76.2 | Unstructured | U |
| GraNet[34] | - | 1.15G | 76.2 | Unstructured | L |
| RigL[15] | 5.1M | 1.68G | 75.1 | Unstructured | L |
| STR[16] | 5.2M | 776M | 76.2 | Unstructured | L |
| GMP[31] | 5.1M | - | 76.5 | Unstructured | L |
| DS(**ours**) | 5.0M | 1.14G | **76.7**(+0.2) | Mixed N:16 | L |
| IncReg[35] | - | 1.37G | 71.1 | Structured | U |
| IE[36] | - | 1.34G | 71.7 | Structured | L |
| Greg-2[22] | - | 1.34G | 73.9 | Structured | L |
| GDP[37] | - | 1.57G | 70.9 | Structured | L |
| FisherPruning[38] | - | 1.02G | 73.9 | Structured | L |
| META[39] | - | 1.00G | 74.4 | Structured | L |
| SR-STE[11] | 6.4M | 1.02G | 76.5 | 4:16 | U |
| SR-STE[11] | 3.2M | 511M | 74.4 | 2:16 | U |
| SR-STE[11] | 1.6M | 256M | 70.7 | 1:16 | U |
| SR-STE[11] | 1.6M | 256M | 71.5 | 2:32 | U |
| Equal model size | | | | | |
| DS(**ours**) | 3.2M | 731M | 75.7(+1.3) | Mixed N:16 | L |
| DS(**ours**) | 1.6M | 392M | 73.5(+2.0) | Mixed N:32 | L |
| Equal FLOPs | | | | | |
| DS(**ours**) | 3.3M | 511M | 75.4 (+1.0) | Mixed N:16 | L |
| DS(**ours**) | 1.7M | 256M | 73.4 (+1.9) | Mixed N:32 | L |

## 4.2 RegNet with FLOPs constraint

We further study our search method on the compact model: RegNet. RegNet [20] is the state-of-the-art compact network architecture searched by NAS from $10^{18}$ candidates. Table 3 shows that layer-wise sparse RegNetX3.2G outperforms its uniform sparse counterpart by 1.0% and dense smaller RegNetX400M by 3.9%, given equivalent FLOPs. The results demonstrate that our method can effectively find the layer-wise sparsity even if the dense model is designed to be compact.

Table 3: RegNets: Comparing sparse RegNets and dense RegNets with various FLOPs. (+) denotes the accuracy gain compared to dense model with same FLOPs.

| Network(Method) | FLOPs | Top1 Acc | Structure |
|---|---|---|---|
| RegNetX400M | 400M | 72.3 | Dense |
| RegNetX800M | 800M | 74.9 | Dense |
| RegNetX1.6G | 1.6G | 76.8 | Dense |
| RegNetX3.2G | 3.2G | 78.2 | Dense |
| Sparse RegNetX3.2G (SR-STE) | 400M | 75.2 | 2:16 |
| Sparse RegNetX3.2G (Ours) | 400M | 76.2(+1.0) (+3.9) | Mixed N:16 |

### 4.3 ResNet56 on CIFAR100

In addition to evaluation performance on the ImageNet dataset, we also evaluate on the CIFAR100 dataset. Results are presented in Table 4. Layer-wise N:M sparsity significantly improves the accuracy on the smaller CIFAR100 dataset under both model-size and FLOPs constraints.

Table 4: ResNet56 on CIFAR100: Comparison between uniform sparsity and layer-wise sparsity. Each setting is evaluated 3 times, mean and std. of top-1 accuracy are reported. Baseline dense ResNet56 obtains 74.8% top-1 acc. with 0.85M parameters, 127M FLOPs.

| Model Complexity | 0.21M#Params | 0.11M#Params | 32M FLOPs | 16M FLOPs |
|---|---|---|---|---|
| Acc. uniform | $72.33_{\pm 0.37}$ | $71.08_{\pm 0.27}$ | $72.33_{\pm 0.37}$ | $71.08_{\pm 0.27}$ |
| Acc. layer-wise | $74.31_{\pm 0.34}$ | $73.26_{\pm 0.24}$ | $74.00_{\pm 0.18}$ | $72.79_{\pm 0.25}$ |
| Acc. gain | +1.98 | +2.18 | +1.67 | +1.71 |

## 5 Ablation study

### 5.1 The advantage of *layer-wise* penalty factor

To study the effectiveness of the ***layer-wise*** penalty factor $\eta_i$ proposed in Section 3.4, we consider a constant penalty factor (1.0) on all layers as the baseline. Table 5 shows that layer-wise schemes searched with $\eta_i$ consistently outperform the counterparts without $\eta_i$. The gap is significant under complexity with small FLOPs (i.e. 16M). To achieve such a low FLOPs without $\eta_i$, both shallow and deep layers are extensively pruned no matter their contributions of FLOPs, resulting in very high sparsity (92.2%). In contrast, with the guidance of $\eta_i$, the sparsity is only 89.0% when achieving equivalent FLOPs. Additional experiments on ImageNet can be found in Supplementary Material.

Table 5: ResNet56 on CIFAR100: Ablation study of layer-wise penalty factor $\eta_i$.

| Model Complexity | 0.21M#Params | 0.11M#Params | 32M FLOPs | 16M FLOPs |
|---|---|---|---|---|
| Acc. with $\eta_i$ | $74.31_{\pm 0.34}$ | $73.26_{\pm 0.24}$ | $74.00_{\pm 0.18}$ | $72.79_{\pm 0.25}$ |
| Acc. w/o $\eta_i$ | $73.80_{\pm 0.20}$ | $72.95_{\pm 0.19}$ | $73.71_{\pm 0.14}$ | $70.39_{\pm 0.19}$ |
| Acc. gain | -0.51 | -0.31 | -0.29 | -2.40 |

### 5.2 Comparing layer-wise N:M to the uniform counterpart using ASP training method

There are two existing methods to train sparse N:M networks: 1) SR-STE[11] and 2) Nvidia ASP [12]. When comparing layer-wise sparse N:M to the uniform baselines in Section 4, we use the most recent SR-STE as the training method to train sparse N:M networks. SR-STE is based on dynamic masking which allows the pruned weights to re-grow during the training/fine-tuning. In contrast, Nvidia ASP is based on fixed masking which determines the mask at the beginning. The mask is fixed and pruned weights are not allowed to re-grow through the fine-tuning stage. Table 6 compares layer-wise sparse N:M networks to the uniform baselines using the ASP method. The results show that layer-wise N:M is consistently better than the uniform counterparts for both ResNet50 and ResNet18, which matches the observations under the SR-STE training setting.

Table 6: Comparing layer-wise to uniform sparsity using the ASP method.

| Network | Structure | #Params | Top-1 Acc(%) |
|---|---|---|---|
| ResNet50 | Uniform 2:16 | 87.5% | 73.3 |
| ResNet50 | Layer-wise N:16 | 87.5% | 74.9(+1.6) |
| ResNet18 | Uniform 2:16 | 87.5% | 64.2 |
| ResNet18 | Layer-wise N:16 | 87.5% | 65.1(+0.9) |

### 5.3 How N:M sparsity achieves comparable results to unstructured sparse sparsity

To study the impact of training settings and how these contribute to the accuracy, we conduct a set of ablation experiments and present results in Table 7. Where **E1** and **E2** are two layer-wise unstructured sparse ResNet50 baselines [16, 31], **E2** - **E7** are ablation experiments which will be explained shortly.

Table 7: Ablation study of the impact of different training settings using ResNet50 with 80% sparsity.

| Index | Structure | Top-1 Acc(%) | Epochs | Training Method | Initialization |
|---|---|---|---|---|---|
| E1 | L unstructured | 76.5 | ~150 | [31] | Random |
| E2 | L unstructured | 76.2 | 100 | STR[16] | Random |
| E3 | L N:M | 76.1 | 100 | SR-STE[11] | Random |
| E4 | L N:M | 76.4 | 120 | SR-STE[11] | Random |
| E5 | L N:M | 76.7 | 120 | SR-STE[11] | Pre-trained |
| E6 | L N:M | 76.6 | 90 | SR-STE[11] | Pre-trained |
| E7 | L N:M | 75.6 | 120 | ASP[12] | Pre-trained |

***Weight initialization:*** In order to recover the accuracy of sparse neural networks, dense weight initialization is typically used [22, 31]. By comparing **E4** to **E5**, it shows that using pre-trained (dense) weight as the initialization helps layer-wise N:M sparse neural networks to reach a higher accuracy.

***Training epochs:*** Training for fine-tuning a sparse neural networks usually requires more epochs for convergence [15]. Experiment pairs **E3** v.s. **E4** and **E5** v.s **E6** in Table 7 demonstrate that longer epochs can help to achieve a higher accuracy, although for settings with dense weight initialization (**E5** vs. **E6**), the accuracy gain of 30 additional epochs is minimal because using dense weights as the initialization can be seen as pre-trained for certain epochs under dense training settings.

***Dynamic vs. Static Mask:*** Experiments **E5** and **E7** compare two different training methods for N:M sparsity. It shows that the dynamic mask sparse training algorithm can outperform the fixed mask training algorithm significantly.

The experimental results in Table 7 indicate that the performance of sparse neural networks can be boosted via pre-trained weight initialization, more training epochs and a proper sparse training method.

## 6 Conclusions

This paper presents an efficient framework, *DominoSearch*, together with the proposed **layer-wise** penalty factor that explicitly quantifies the layer-wise computation complexity and parameter redundancy, to find the layer-wise fine-grained N:M sparsity from dense weights.

We experimentally validate the advantages of the layer-wise N:M sparsity on the ImageNet and CIFAR100 datasets with various networks. The results of ResNet50 on ImageNet show that layer-wise N:M sparsity is competitive to layer-wise unstructured sparsity which is believed to be the upper bound of neural pruning with respect to the accuracy-sparsity trade-off. We hope that our contributions can encourage and draw attentions to flexible fine-grained N:M hardware-software system design/research.

## 7 Limitations

The major limitation of our work is that we did not benchmark deployment gains such as latency and throughput. To the best of our knowledge, Nvidia Ampere Tensor Core is the only commercially available architecture that supports fine-grained N:M sparse computation efficiently. However, it only supports limited options for fixed sparsity (2:4 for bfloat and 4:8 for integer). Thus, following the common practice of related work we compared to, we used theoretical computational complexity, FLOPs, to estimate the computation performance gains.

## Acknowledgments and Disclosure of Funding

We thanks to all anonymous reviewers for their feedback on improving this work. This work is funded by the Dutch Research Council (NWO) Perspectief program ZERO-ARM P3. We thanks for the computation resources provided by Sensetime Research and CUHK-Sensetime Joint Lab.

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
