# DominoSearch: Find layer-wise fine-grained N:M sparse schemes from dense neural networks – Supplementary Material

Organization of Supplementary Material:

- Section 1: Additional ablation experiments on ImageNet dataset for proposed layer-wise penalty factor.
- Section 2: Experimental study of a different policy with fixed N and flexible M.
- Section 3: Sensitivity of hyper-parameter $\beta_1$ and $\beta_2$ in proposed layer-wise penalty factor.
- Section 4: Detailed experimental settings including datasets and hyper-parameters used in main paper.

## 1 Additional ablation experiments on ImageNet dataset

In addition to the ablation experiments on the CIFAR100 dataset, we further evaluate the effectiveness of the layer-wise penalty factor $\eta_i$ on the large-scale ImageNet dataset using the ResNet18 architecture. Table 1 shows results when using a model size constraint and 2 presents results when using the FLOPs constraint. The results confirm that the layer-wise sparse schemes searched with $\eta_i$ consistently outperform the schemes without $\eta_i$ under both the model-size constraint and the FLOPs constraint.

Table 1: ResNet18 - model size constraint

|  | Params | Top1 Acc | Structure |
|---|---|---|---|
| with $\eta_i$ | 1.46M | 68.76 | Mix N:16 |
| w/o $\eta_i$ | 1.46M | 68.24(-0.52) | Mix N:16 |

Table 2: ResNet18 - FLOPs constraint

|  | FLOPs | Top1 Acc | Structure |
|---|---|---|---|
| with $\eta_i$ | 227M | 67.98 | Mix N:16 |
| w/o $\eta_i$ | 227M | 67.07(-0.92) | Mix N:16 |

## 2 Study on the policy of fixed N

In the main paper, we assume a policy with fixed M and flexible N. Furthermore, we also use a design space with N equal to a power-of-two. In this section, we extend these experiments with an additional policy with fixed N and flexible M. This is achieved by transforming the schemes of fixed M. For instance, 8:16, 4:16, 2:16 and 1:16 will be transformed as 1:2, 1:4, 1:8 and 1:16 with fixed N (1) and flexible M (2,4,8,16). Results are shown in Table 3. As can be seen, the policy with fixed M consistently outperforms the policy of fixed N. A possible explanation is that larger M results in more flexibility when selecting important weights, given the same sparsity percentage. Figure 1 and 2 illustrate the differences between 1:2 and 2:4 with the same dense weight matrix and sparsity (i.e. 50%). After pruning, the $L_1$ norm of the 2:4 sparse weight matrix is **0.5542** which is larger than **0.5094** (i.e., the $L_1$ norm of the 1:2 sparse weight matrix). This indicates that a sparsity pattern with larger M (in this example, 4) can keep more important weights [1] than smaller M (e.g., 2). From the results in Table 3 we can conclude that for layer-wise schemes searched by DonimoSearch, a policy with fixed M is better.

Preprint. Under review.

Table 3: Performance for fixed N and fixed M policies using ResNet56 on CIFAR100.

| Model Complexity | 0.21M#Params | 0.11M#Params | 32M FLOPs | 16M FLOPs |
|---|---|---|---|---|
| Acc. uniform | $72.33_{\pm 0.37}$ | $71.08_{\pm 0.27}$ | $72.33_{\pm 0.37}$ | $71.08_{\pm 0.27}$ |
| Fixed M | $74.31_{\pm 0.34}$ | $73.26_{\pm 0.24}$ | $74.00_{\pm 0.18}$ | $72.79_{\pm 0.25}$ |
| Fixed N | $73.86_{\pm 0.26}$ | $72.48_{\pm 0.28}$ | $73.72_{\pm 0.21}$ | $71.98_{\pm 0.17}$ |
| Acc. drop | -0.45 | -0.78 | -0.28 | -0.81 |

$$\mathbf{w}\begin{bmatrix} 0.0104 & 0.0114 \\ 0.0020 & 0.0061 \\ 0.0212 & 0.0748 \\ 0.0368 & 0.0898 \\ 0.0854 & 0.1751 \\ 0.0406 & 0.0450 \\ 0.0896 & 0.0169 \\ 0. & 0.0177 \end{bmatrix} -> \widetilde{\mathbf{w}}\begin{bmatrix} 0. & 0.0114 \\ 0. & 0.0061 \\ 0. & 0.0748 \\ 0. & 0.0898 \\ 0. & 0.1751 \\ 0. & 0.0450 \\ 0.0896 & 0. \\ 0. & 0.0177 \end{bmatrix} \overset{L_1}{->} [0.5094]$$

Figure 1: Sparsity **1:2**. For every two elements, one will be removed based on the magnitude criterion. Values in blue represent the values kept in the dense weight matrix **w**. With 1:2 sparsity, the $L_1$ norm of the sparse weight $\widetilde{\mathbf{w}}$ is **0.5094**.

# 3 Sensitivity of hyper-parameter $\beta_1$ and $\beta_2$

We further study the sensitivity of $\beta_1$ and $\beta_2$ of Equation 1.

$$\eta_i = \beta_1 * c_i + \beta_2 * r_i. \tag{1}$$

Where $c_i$ is the layer-wise computational complexity factor and $r_i$ is the layer-wise redundancy factor. Details can be found in Section 3.4 of the main paper.

Table 4: Effect of different $\beta_1$ and $\beta_2$ using ResNet56 on CIFAR100.

| Model Complexity | 0.21M#Params | 0.11M#Params | 32M FLOPs | 16M FLOPs |
|---|---|---|---|---|
| Acc. uniform | $72.33_{\pm 0.37}$ | $71.08_{\pm 0.27}$ | $72.33_{\pm 0.37}$ | $71.08_{\pm 0.27}$ |
| FLOPs/Model size | 32M | 16M | 0.21M | 0.11M |
| $\beta_1 = 0.5, \beta_2 = 0.5$ | | | | |
| Acc. Equal model size | $74.31_{\pm 0.34}$ | $73.26_{\pm 0.24}$ | - | - |
| FLOPs | 43.3M | 25.6M | | |
| $\beta_1 = 0.0, \beta_2 = 1.0$ | | | | |
| Acc. Equal model size | $74.20_{\pm 0.35}$ | $73.17_{\pm 0.25}$ | - | - |
| FLOPs | 53.4M | 26.0M | | |
| $\beta_1 = 0.8, \beta_2 = 0.2$ | | | | |
| Acc. Equal FLOPs | - | - | $74.00_{\pm 0.18}$ | $72.79_{\pm 0.25}$ |
| Model size | - | - | 0.19M | 0.09M |
| $\beta_1 = 1.0, \beta_2 = 0.0$ | | | | |
| Acc. Equal FLOPs | - | - | $73.97_{\pm 0.26}$ | $72.80_{\pm 0.18}$ |
| Model size | - | - | 0.20M | 0.11M |

Table 4 shows the results with different $\beta_1$ and $\beta_2$ under both a model-size constraint and a FLOPs constraint. The effect of $\beta_1$ and $\beta_2$ on accuracy is negligible but they have a noticble effect on the

$$
\begin{bmatrix}
0.0104 & 0.0114 & 0.0020 & 0.0061 \\
0.0212 & 0.0748 & 0.0368 & 0.0898 \\
0.0854 & 0.1751 & 0.0406 & 0.0450 \\
0.0896 & 0.0169 & 0. & 0.0177
\end{bmatrix}
\overset{\mathbf{w}}{->}
\begin{bmatrix}
0.0104 & 0.0114 & 0. & 0. \\
0. & 0.0748 & 0. & 0.0898 \\
0.0854 & 0.1751 & 0. & 0. \\
0.0896 & 0. & 0. & 0.0177
\end{bmatrix}
\overset{\widetilde{\mathbf{w}}}{} \overset{L_1}{->} [0.5542]
$$

Figure 2: Sparsity **2:4**. For every four elements, two will be removed based on the magnitude criterion. Values in blue represent the values kept in the dense weight matrix $\mathbf{w}$. With 2:4 sparsity, the $L_1$ norm of the sparse weight $\widetilde{\mathbf{w}}$ is **0.5542**.

second complexity metric (i.e. **FLOPs** when using model size as constraint and vice-versa **model size** is affected when using FLOPs as a constraint). For instance, when comparing ($\beta_1 = 0.5, \beta_2 = 0.5$) and ($\beta_1 = 0.0, \beta_2 = 1.0$), the layer-wise sparse models' top1 accuracy are 74.31 and 74.20 respectively with the same model size (0.21M). The relative change is only $|\frac{74.31-74.20}{74.20}| * 100\% = 0.14\%$. On the other hand, the second complexity metric FLOPs of these two models are 43.4M and 53.4M respectively, which corresponds to a relative change of $|\frac{43.3-53.4}{53.4}| * 100\% = 18.9\%$.

# 4 Detailed Experimental settings

**Dataset** ImageNet-1K [2] is a large-scale image classification task, known as one of the most challenging image classification benchmarks. It consists of more than 1.2 million training images and 50K validation images with a size of 224x224 pixels. Each image is labelled as one of 1K classes. CIFAR100 [3] is a smaller-scale image classification dataset consisting of 100 classes. Each class has 500 training colour images and 100 testing images of 32x32 pixels in size.

We first apply DominoSearch to search the layer-wise schemes from pre-trained dense models. For ResNet18/50 [4], the pre-trained models have been downloaded from Pytorch Model zoo[1]. For RegNetX3.2G [5], the dense pre-trained model has been downloaded from the official Githup repository[2], provided by the authors. For ResNet56 on Cifar100, we train the dense model ourselves and then apply DominoSearch.

## 4.1 DominoSearch

Table 5 shows the hyper-parameter settings of DominoSearch. The explanations of these hyper parameters can be found in Section 3 of the main paper.

Table 5: Hyper Parameter settings for DominoSearch.

| | |
|---|---|
| Solver | SGD |
| SGD weight decay | 0.1 |
| $k_p$ | 5 |
| $K_c$ | 100 |
| $v_r$ | 0.75 |
| LR policy | 0.01(fixed) |
| Batch size | 64 |

The search phase takes 2-4 hours when searching on ImageNet with a single RTX2080Ti GPU, depending on network and complexity constraints.

## 4.2 ImageNet Experiments

With searched layer-wise schemes, we retrain the sparse networks with pre-trained weights as initialization to recover the accuracy. To train layer-wise fine-grained N:M sparse models, we adapt the setting of the SR-STE paper [6]. Table 6 shows the applied hyper parameters for reproduction.

---

[1]`https://pytorch.org/vision/stable/models.html`
[2]`https://github.com/facebookresearch/pycls`

Table 6: Hyper Parameter settings for on ImageNet.

| Solver | SGD(0.9,5e-4) |
|---|---|
| $\lambda$ [6] | 0.0005 |
| LR policy | cosine,base_lr=0.01 |
| Batch size | 256 |
| Epochs | 120 |

For ResNet50, the training phase takes $\sim 40$ hours on 8 RTX2080Ti GPUs.

## 4.3 CIFAR100 Experiments

We train the dense model with hyper parameters as presented in Table 7.

Table 7: Hyper Parameters for training the dense model on CIFAR100.

| Solver | SGD(0.9,1e-4) |
|---|---|
| LR policy | Multi-step (0:0.1,80:0.01,120:0.001 ) |
| Batch size | 64 |
| Epochs | 160 |

Table 8 shows the hyper parameters for training layer-wise N:M sparse model.

Table 8: Hyper Parameter for training the sparse model on CIFAR100.

| Solver | SGD(0.9,5e-4) |
|---|---|
| $\lambda$ [6] | 0.0005 |
| LR policy | Multi-step (0:1e-2,80:1e-3,120:1e-4 ) |
| Batch size | 64 |
| Epochs | 160 |