# OpenReview forum: "DominoSearch: Find layer-wise fine-grained N:M sparse schemes from dense neural networks"
_NeurIPS.cc/2021/Conference — NeurIPS 2021 Poster_

### Official Review · Reviewer_6z8t · 2021-06-29

**Rating:** 7
**Confidence:** 5

**Summary:**

The abstract and title of the paper convey the summary accurately. This paper proposes to improve the recently proposed work on N:M sparsity by Zhou et al., ICLR 2021. The former work results in models that have N:M unstructured sparsity that can be used in newer accelerators like Ampere but are handicapped by the fact that all the layers have the same sparsity ratios. This results in the inability to optimize for FLOPs for a given # parameter count thus having a knob for tuning the tradeoffs.

This paper proposes to fix that by proposing an iterative scheme called DominoSearch to find layer-wise fine-grained schemes/rations for N:M sparsity providing much-needed flexibility. DominoSearch operates on pre-trained models with a budget constraint and overtime figures out the good-enough mixed N:M scheme along with layer-wise sparsity for all the weight tensors on the network (layers). This is achieved with weight penalty as well as explicit layer-wise penalties combined with standard group-wise sparsity used for N:M scheme. The layer-wise penalty factor helps in trading off accuracy, FLOPs and model size is based on the sparsity achieved iteratively + a prefixed scheme like ERK for measuring redundancy.

With one caveat that all the baselines (except Zhou et al ICLR 2021) are trained from scratch and sparsity is induced during that time, while this paper works on pre-trained models. The experimental section is super strong and has strong practical implications along with a clear Pareto dominance on the baselines.

**Limitations And Societal Impact:**

Mentioned above. The authors need to talk about training overhead and sort of unfair comparison by using pre-trained models for N:M schemes.

**Main Review:**

The brevity of the review should not be taken as a negative for this paper as I have very few things to comment on and the paper is very clear on what it is trying to do. I think I summarized the strengths in the summary but will try to cover if there any I missed. I have a couple of concerns/comments that will help me understand and place the empirical evaluation in context while also improving the paper even further. I will be going sequentially.

Strengths:
1) The motivation is super clear. N:M sparsity is useful, but uniform layer-wise nature is restrictive, so figure out the mixed scheme with varied N:M along with varying layer-wise sparsity to tune for the tradeoff if needed and further increase accuracy for fixed resource costs compared to the uniform scheme.
2) Related work is extensive and puts things in context.
3) Alg 3.1 is clean and helps understand the scheme well.
4) I like the ERK-based redundancy estimate in Eq 8, that is novel as far as I know and can also be used for RL-based search if needed.
5) The practical implications of the results, be it constant FLOPs or constant params are impressive coupled with the accuracy gains.
6) extensive experimentation and the ablations should be commended.
7) The authors also do a good job in addressing limitations, except one thing which I will mention later.

Weaknesses:
Please include a short primer on N:M sparsity if possible to have a sense of completeness to the reader.
1) The abstract can be condensed and also the claim that STR being the upper bound for compression accuracy should be removed (that would be an overarching claim).
2) The math in the method section for group-wise threshold should be made much clearer and the same goes with the weight penalty. The layer-wise penalty is very clean and would expect something of that state for the other two. It is slightly hard to parse through in the current form.
3) I think I have only one concern and it is kind of a major one, even though it doesn't take away the benefits this method brings to the table.
I am not completely clear on why this method can't be applied for training from scratch. Even with the pre-trained model, after the search, the retraining still needs 120 epochs for ImageNet which is even more than the normal training of imageNet. The training costs are something that isn't clearly discussed and do fit into limitations as this is a very expensive scheme, maybe better than RL, but still an expensive scheme to get to the final model via the retraining phase. I would like the authors to comment on these issues 1) training overheard, 2) why not train from scratch and why do we need the pre-trained models. Lastly, a simple RL-based scheme would be a strong addition to the baselines which will do the same kind of search but show why the greedy approach proposed in much more efficient.

Addressing these concerns will further improve the paper and will lead me to improve the rating. Even though the training overhead is a lot, I believe the final models are worth it, however, if this can be done during training (not on pre-trained models) that also improves the training overhead and other things involved.

**Time Spent Reviewing:**

3 hrs

---

> ### Author Response · Authors · 2021-08-10
> **Responses to Reviewer 6z8t**
>
> Thanks for your constructive feedback and your appreciation to our work. We respond to your concerns as following:
>
>
> **Q1**: The abstract can be condensed and also the claim that STR being the upper bound for compression accuracy should be removed (that would be an overarching claim).
>
>
> **A1**: Thanks for the advice. We will squeeze the abstract and remove the claim about STR in revised submission.
>
> **Q2**: The math in the method section for group-wise threshold should be made much clearer and the same goes with the weight penalty.
>
> **A2**: Thanks for the advice. We will use better math symbols add some  explanations to give a clearer expression.
>
> **Q3**: 1)training overhead
>
> **A3**: SRT-STE uses 120 epochs for training, we follow the same setting to recover the accuracy of the sparse network. Indeed, new experiment suggests that the number of epochs can be actually reduced to 90 if we use dense weights as initialization. We did not study the training cost in our first submission because we did not focus on training algorithm, but we will add this kind of discussion in revised submission.
>
>
> | Index |  Top-1 Acc (%) |  Epochs|
> |-|:-:|:-:|
> | ResNet50(80% Sparsity)  |  76.7  |  120  |
> | ResNet50(80% Sparsity) |  76.6  |    90  |
>
>
>
> **Q4**: 2) why not train from scratch and why do we need the pre-trained models.
>
>
> **A4**: In our experiments setting, we search with dense weights because after training the **weight distributions of each layer** are **stable and accurate enough**. It is a straightforward choice that we train our sparse model from dense weights initialization, because we search layer-wised sparse ratio (N) from dense pre-trained model. In order to study the necessity of searching with dense weights, we conduct additional experiments with steps: 1) pre-train model with x epochs. 2) apply search algorithm to find layer-wise sparsity 3) fine-tune with the searched sparsity to recover accuracy.
>
> Table below shows the results.
>
> |index |  Top-1 Acc (%) | Params(Sparsity)  | FLOPs | pre-trained epochs|
> |-|:-:|:-:|:-:|:-:|
> | 1 |  76.2     |     5.0M (80%)  |  1.14G   | 0(random init)   |
> | 2 |  76.3      |     5.0M (80%)  |  1.14G   | 20   |
> | 3 |  76.7      |     5.0M (80%)  |  1.14G   | 90(Pytorch model zoo)   |
>
>
> The results(1,2,3) show that searching with dense weights gives higher model accuracy. Although we believe that the epochs can be reduced and we do not need fully trained weights (i.e. we do not need to pre-train 90 epochs to get stable weight distributions, fewer epochs could be sufficient), we could not study this in details due to limited time. We will add this discussion in the revised submission.
>
> **Q5**: a simple RL-based scheme would be a strong addition to the baselines which will do the same kind of search but show why the greedy approach proposed in much more efficient.
>
> **A5**: We believe RL-based search algorithm is promising because it has been used for mixed-bit quantization search problem [2]. This paper focuses on simple and efficient search algorithm and avoids time-consuming stages. We believe that RL-based search algorithm is able find better layer-wise sparsity schemes at the cost of longer search stage, it would be an interesting future work to explore RL-based algorithm and see the cost it has to pay and gains we can get.
>
>
> **Q6**: if this can be done during training (not on pre-trained models) that also improves the training overhead and other things involved.
>
> **A6**: As we discussed in Q4, the re-training cost can be reduced (at least from 120 epochs to 90 epochs). The current search algorithm is a two-stage search algorithm which requires re-training after searching to recover the accuracy (e.g. [3]). We believe one-stage algorithm such as weight-sharing approaches [4,6] is interesting future work.
>
>
> [1]. Frankle, J., Dziugaite, G. K., Roy, D. M., & Carbin, M. (2020). Pruning neural networks at initialization: Why are we missing the mark?. arXiv preprint arXiv:2009.08576 (ICLR2021).
>
> [2]. HAQ: Hardware-Aware Automated Quantization with Mixed Precision. Kuan Wang et al. CVPR2019.
>
> [3] Rethinking Differentiable Search for Mixed-Precision Neural Networks. Zhaowei Cai et al. CVPR2020.
>
> [4] BigNAS: Scaling Up Neural Architecture Search with Big Single-Stage Models. Jiahui Yu et al. ECCV2020
>
> [6] Once-for-All: Train One Network and Specialize it for Efficient Deployment. Han Cai et al. ICLR2020
>
> [15] Neural Pruning via Growing Regularization, Huan Wang, et al. ICLR 2021.
>
> *Hope our above responses are helpful to address your concerns. If you have further questions, please let us know. Thanks!*

---

> > ### Comment · Reviewer_6z8t · 2021-08-14
> > **Thanks for the response. Will get back.**
> >
> > Dear Authors,
> >
> > Thanks for the detailed response. I am traveling right now and will try to get back on this in a week or so. Apologies for the delay.

---

> > ### Comment · Reviewer_6z8t · 2021-08-15
> > **Thanks for the response**
> >
> > Dear Authors,
> >
> > Thanks for the detailed response. I am very much looking forward to these changes in the paper. Especially the table and ablations included in the rebuttal for Q3 and Q4.
> >
> > I vote for acceptance.

---

> > > ### Author Response · Authors · 2021-08-16
> > > **Thanks for you feedback and appreciation**
> > >
> > > Dear Reviewer:
> > >
> > > Thanks for your appreciation to our work and constructive reviews. We will include the experiments and discussions in the revised version.
> > >
> > > Kind Regards,
> > > Paper4078 Authors.

---

### Official Review · Reviewer_avap · 2021-07-11

**Rating:** 6
**Confidence:** 3

**Summary:**

This work proposed a training algorithm that allows models to use different sparity levels in different network layers under the N:M sparsity setting.

**Limitations And Societal Impact:**

Yes

**Main Review:**

This paper tested the scheme that allows network layers in a single model to have different sparsity levels under the N:M sparsity setting. Their experiment results clearly show that their proposed scheme is working. However, the results are well expected given what we have observed in the earlier works. Therefore, the significance of this work is limited. I also suggest the authors improve the clarity of the manuscript, especially the method section. Please also see the following detailed comments.

Detailed comments:

P3,L89: N:M sparsity was originally proposed in ref1(called intragroup sparsity there). Nvidia first made the idea working on the real hardware. I think ref1 still deserves some credit.

Fig 3: the last row of T^i should be 0.0089 instead of 0.0085

P4: Section on "group-wise threshold selection": The reasoning in this part is very confusing, please try to clean it up.

P5, L161: why the L_2 norm can hold a group-wise feature?

Equation (5): k_p mentioned in L165 but not clearly defined, Seems it is switch to K_c

Suggestion: There are too many different N defined, maybe try to avoid overload it.

Algorithm 3.1: t is always 0 according to the pseudo-code. v_r and the voting process not clearly described. L8: The >= should be reversed.

It appears that the word sparsity has been used to describe various notions, such as sparsity level, sparsity pattern. Please clear them up.


Table 2, the accuracy rate for the dense ResNet50 is much lower than the one provided in the original ResNet paper

ref1:
Wu X, Liu X, Li W & Wu Q. Improved Expressivity Through Dendritic Neural Networks. NeurIPS 2018




**Time Spent Reviewing:**

4

---

> ### Author Response · Authors · 2021-08-10
> **Responses to Reviewer avap**
>
>
> Thanks for your constructive comments to help us improve the quality of the work. We respond to your concerns as following:
>
> **Q1**: N:M sparsity was originally proposed in ref1 (called intragroup sparsity there). Nvidia first made the idea working on the real hardware. I think ref1 still deserves some credit.
>
> **A1**: Thanks for pointing out this paper, we will include it in introduction section in revised submission.
>
> **Q2**: Section on "group-wise threshold selection": The reasoning in this part is very confusing, please try to clean it up.
>
> **A2**: We will re-organize the expression more clear way in revised submission. For instance, to draw an abstract figure to help readers to understand the key concepts.
>
> **Q3**: why the L_2 norm can hold a group-wise feature?
>
> **A3**: The motivation of using L_2 norm(a/2 * sqrt(w^2)) is that after L_2 norm will apply the magnitude of weights (a*|w|) as penalty to each layer during backpropagation. It fits the magnitude-based threshold and criteria for each group. We will add more explanations in revised submission.
>
> **Q4**: Equation (5): k_p mentioned in L165 but not clearly defined, Seems it is switch to K_c
>
> **A4**:  $k_p$ is the interval to apply weight penalty (L2 norm), which we follows the strategy in [15]. K_C is the interval to check the threshold and remaining weights to determine if current N should be decreased (i.e., increasing sparsity). We will add more explanations to avoid confusion in revised submission.
>
> **Q5**: There are too many different N defined, maybe try to avoid overload it.
>
> **A5**: Thanks for the advice. We will use clearer definitions in revised submission to avoid confusion.
>
> **Q6**: Algorithm 3.1: t is always 0 according to the pseudo-code. v_r and the voting process not clearly described. L8: The >= should be reversed.
>
> **A6**: Thanks for pointing out these issues. Voting process is described in Line 145-147. We will add more explicit explainations in revised submission.
>
> **Q7**: It appears that the word sparsity has been used to describe various notions, such as sparsity level, sparsity pattern. Please clear them up.
>
> **A7**: Thanks for pointing out this issue. We will use consistent  terminology to avoid confusion.
>
> **Q8**: Table 2, the accuracy rate for the dense ResNet50 is much lower than the one provided in the original ResNet paper
>
> **A8** We agree that it is not clearly explained in our initial submission. The pre-trained dense models ( downloaded directly from Pytorch Model Zoo[1]) are the model we start searching. It serves as a reference for dense parameters and FLOPS, we will use a better dense baseline accuracy(77.3%).
>
> [1] Pytorch Model Zoo. https://pytorch.org/vision/stable/models.html
>
> [15] Neural Pruning via Growing Regularization, Huan Wang, et al. ICLR 2021.
>
> *Hope our above responses are helpful to address your concerns. If you have further questions, please let us know. Thanks!*

---

> > ### Comment · Reviewer_avap · 2021-08-24
> > **Thanks for the response**
> >
> > Thank you for the response and your extensive discussion with reviewer Mirr. Your discussion with reviewer Mirr also addressed my main concern on the baseline used for comparison as in Q8. Therefore, I have raised my rating to 6. Please update your paper to reflect the results of the discussion. I also agree an M size of 32 is probably too large. I suggest limiting the M size below 16 in the final version of the paper.

---

> > > ### Author Response · Authors · 2021-08-25
> > > **Thanks for your reviews and appreciation**
> > >
> > > We are happy that our responses to Reviewer-Mirr can help to address your concerns. Thanks for your reviews and appreciation to our new experiments/discussions included in our previous responses.
> > >
> > > Kind Regards,
> > >
> > > Paper4078 Authors.

---

### Official Review · Reviewer_Mirr · 2021-07-15

**Rating:** 6
**Confidence:** 5

**Summary:**

The authors propose combining two existing techniques, N:M sparsity and per-layer (rather than uniform/global) pruning rates.  By allowing each layer to choose a particular N:M pair, the network can find a configuration that is not constrained to sharing a single N:M.  The solution is DominoSearch, an algorithm which starts with pretrained (dense) weights and determines a pruning threshold for each group of M weights based on the initial values.  Some number of iterations of fine-tuning adjust the weights gradually lower, in turn reducing each layer's chosen N.  Once some complexity target (either parameter or FLOP count) is hit, the network's layer-wise N:M are frozen, and the weights are finally fine-tuned.  A layer-wise penalty factor adjusts the overall complexity metric to account for layers which have relatively more or fewer FLOPs.  Empirical results show compelling reductions in FLOPs and parameters when compared to training with unstructured sparsity or uniform N:M sparsity.

**Limitations And Societal Impact:**

Yes

**Main Review:**

## Originality

The idea to start with a high N that drops over time is akin to iterative magnitude pruning for unstructured sparsity, but I haven't seen a principled approach for N:M sparsity that doesn't rely on sweeping hyperparameters, etc.  Similarly, pruning different layers to different amounts is nothing new, but it is when applied to N:M structured sparsity.

## Quality

I'm concerned that the comparisons used to make claims like "… [the proposed result] is competitive to accuracy of the state-of-the-art layer-wise unstructured-sparsity model" aren't fair.  Particularly, the problem being solved: DominoSearch is an algorithm to choose N:M values per layer of an already-trained dense network, but all comparisons are to algorithms which induce sparsity "from scratch," i.e. during the initial training process, without first training a dense network.  Looking instead at a result for 80% sparsity (roughly matching the 25.6->5 Mparams result from Table 2) from Figure 5 in [1], the accuracy is 76.52%, barely reduced from the baseline of 76.69%.  (This result is two years old, so the SOTA may have advanced since then.)  Does it make sense that accuracy for structured sparsity would be higher than accuracy for unstructured sparsity?  If this is the case, then is it due to regularization, suggesting that the dense model was poorly trained to begin with?  If, as you point out, unstructured sparsity is believed to be the upper bound, but you find that it isn't, I'd expect some discussion of why this commonly-held belief is not true.

Similarly, for the uniform N:M case, the comparison seems to be against SR-STE, which is a method to find N:M masks during training (with no fine-tuning).  Instead, a better comparison would be to use the method implemented in the submission's [6] (or, the more recent whitepaper[2], which may be a better source than [6] and [4] (which is not titled appropriately in the bibliography)), which uses the same "start with a dense model, determine a mask, and fine-tune while enforcing the mask" process as this submission.

If we set aside the comparisons to inappropriate baselines, the algorithm does seem to work as advertised: the ability to optimize for either FLOPs or number of parameters is great, and Table 4 suggests that for a small data set, layer-wise N:M is superior to uniform N:M at a given FLOPs or parameter target.  (Unless, of course, these results also use SR-STE for the "uniform" representation; it's not clear from the Table, but all other results do use SR-STE.)   Further, there's no catastrophic accuracy loss, so the resulting models are plausibly useful, particularly when compared to a dense model with equivalent complexity.

A comparison of the layer-wise N:M vs. uniform N:M at the same accuracy (as dense) would be useful.  For example, from the results in Table 3, the user must make hard decisions.  Would it be better to start with RegNetX1.6 and prune by an overall factor of 4x (FLOPs) to reach the 400MFLOPs target?  If accuracy were the same as the unpruned RegNetX3.2, the choice is easy - there's no downside to pruning RegNetX3.2, and RegNetX1.6 cannot reach that accuracy.  Existing results for uniform 2:4 sparsity show that accuracy is maintained (at 50% parameters), so if DominoSearch cannot achieve this same accuracy/sparsity target, there will be obvious room for improvement; if it is able to improve upon the parameter reduction while maintaining accuracy, then its superiority will be obvious.

## Clarity
The submission is generally well-written.  There were some details that should be included in the main text, though:
- How much fine-tuning is utilized after the layer-wise N:M search completes?
- How is kp chosen?

The hyperparameter kp is confusing in the presence of 'k' as the group index.

I have to admit that I don't follow the motivation for the name, even with the description on lines 59-60: "As indicated by the name Domino, the N value from the N:M ratio decrements during the search phase with the weight penalty acting as momentum."

Figure 4(c) shows the parameter distribution for which search target?

GDP and META are missing units in the FLOPs column in Table 2.

## Significance
This work won't lead to an immediate performance speedup, but that's just fine.  Instead, it seeks to guide future N:M sparse architecture designs.  If the results hold up after comparing to reasonable baselines (prune then finetune, rather than sparse from scratch), then hardware that supports variable N:M may be well-motivated.

However, some of the N:M settings used in this submission may be on the extreme side.  If 2:4 is what is supported today, extending to M=32 may not be reasonable from a hardware complexity standpoint.  Reducing the size of M and showing compelling results would be a stronger argument for supporting such a design.

---

[1] "The State of Sparsity in Deep Neural Networks," Gale et al., https://arxiv.org/abs/1902.09574

[2] "Accelerating Sparse Deep Neural Networks," Mishra et al., https://arxiv.org/abs/2104.08378

**Time Spent Reviewing:**

4

---

> ### Author Response · Authors · 2021-08-10
> **Responses to Reviewer Mirr, Part 2**
>
>
> **Q4**: **Quality:** the comparison seems to be against SR-STE, which is a method to find N:M masks during training (with no fine-tuning). Instead, a better comparison would be to use the method implemented in the submission's [6] (or, the more recent whitepaper[2], which may be a better source than [6] and [4] (which is not titled appropriately in the bibliography)), which uses the same "start with a dense model, determine a mask, and fine-tune while enforcing the mask" process as this submission.
>
> **A4**: Thanks for the remark, we agree this comparison could be useful. However, we want to emphasize that the contribution of this paper is the search algorithm. Table 2 in [5] has demonstrated that SR-STE is a better training algorithm for N:M sparsity than Nvidia ASP, so we chose SR-STE training settings as our comparison and baseline training. It is a natural choice that we train our model from dense initialization, because we search layer-wised sparse ratio (N) from a **dense pre-trained model**. We have added experiment with random initialization for fair comparison. The results is comparable 76.4 (**random** initialization) vs 76.7 (**dense** initialization) as shown the table of A2.  We will correct the title in the bibliography in revised submission.
>
> --------------------------------
> **Q5**: **Quality:** A comparison of the layer-wise N:M vs. uniform N:M at the same accuracy (as dense) would be useful. For example, from the results in Table 3, the user must make hard decisions. Would it be better to start with RegNetX1.6 and prune by an overall factor of 4x (FLOPs) to reach the 400MFLOPs target? If accuracy were the same as the unpruned RegNetX3.2, the choice is easy - there's no downside to pruning RegNetX3.2, and RegNetX1.6 cannot reach that accuracy. Existing results for uniform 2:4 sparsity show that accuracy is maintained (at 50% parameters), so if DominoSearch cannot achieve this same accuracy/sparsity target, there will be obvious room for improvement; if it is able to improve upon the parameter reduction while maintaining accuracy,
>
> **A5**: Sorry that we do not fully understand the question
> > If accuracy were the same as the unpruned RegNetX3.2
>
> So we try our best to respond reviewer's concern based on our interpretation.
>
> 1) We add the result of RegNetX1.6G with uniform 4:16 sparsity(400MFLOPs complexity). The results below show that 4:16 RegNetX1.6G achieves higher accuracy than 2:16 RegNetX3.2G, and lower than mixed N:16 RegNetX3.2G.
> Sparsification will lead to **significant accuracy drops** after a certain sparsity threshold. This sparsity threshold for ResNet50 is around 90% (as indicated in Figure 5 of [24]). For compact RegNetX this number could be smaller (e.g. 80% or 70%). This can explain why uniform 4:16(75%) RegNetX1.6G outperforms (or very close to) uniform 2:16(87.5%) RegNetX3.2G given same FLOPS(400M) complexity. Note that the network architecture of RegNetX1.6G and RegNetX3.2G (and other RegNetX variants) is searched via NAS **independently**. The architecture of RegNetX3.2G is **not** gained by simply doubling the channels of RegNetX1.6G, so it is difficult to make direct comparisons between their sparse(sub)networks.
> This observation could trigger the possible discussions about which dense network ( e.g. RegNetX1.6G v.s. RegNetX3.2G ) we should compress from given a complexity target, but this is out of our submission's scope.
>
>    | Model | FLOPs | TOP1 | Structure |
>    | -------- | -------- | -------- | -----|
>    | RegNetX400M     | 400M     | 72.3     | Dense  |
>    | RegNetX1.6G     | 400M     | 75.4     | 4:16 Sparse  |
>    | RegNetX3.2G     | 400M     | 75.2     | 2:16 Sparse  |
>    | RegNetX3.2G     | 400M     | **76.2(+3.9)**     | Mixed N:16 Sparse |
>    | RegNetX1.6G     | 1.6G     | 76.8     | Dense  |
>    | RegNetX3.2G     | 400M     | 78.2     | Dense |
>
> 2) We would like to emphasize that the purpose of Table 3 (of our submission) is to investigate if layer-wise N:M sparsity can be compatible with **compact dense models searched by NAS**. RegNetX400M can be seen as (dense)subnetwork of RegNet3.2G, and sparse RegNetX3.2G with 400M FLOPs is also a (sparse)subnetwork of RegNet3.2G. The (sparse)subnetwork of RegNet3.2G significantly outperforms RegNet400M by about **3.9%**. This observation shows that layer-wise N:M will be an important component to reduce complexity given the fact that **1)** fine-grained structured sparsity can be supported by hardware efficiently **2)** the layer-wise scheme search cost is more affordable than NAS.
>
> ------------------
>
> **Q7**: **Clarity:** How much fine-tuning is utilized after the layer-wise N:M search completes?
>
> **A7**: We adopt training settings of SR-STE [5]. The difference is that we use pre-trained dense weights (from Pytorch model zoo) for initialization because we complete search with dense weights. For fine-tuning we use same epochs 120 as [5]. We learn this accuracy recovery strategy (i.e. same epochs as training dense model) from [15].
>
>
> **Q8**: **Clarity:**  How is kp chosen?
>
> **A8** Kp is used as buffer time for weights to adjust for applied penalty, we follow the strategy in [15].
>
> **Q9**: **Clarity:**  The hyperparameter kp is confusing in the presence of 'k' as the group index.
>
> **A9** Thanks for the advise, we will use a different symbol to avoid confusion in revised submission
>
> **Q10**: **Clarity:**  I have to admit that I don't follow the motivation for the name, even with the description on lines 59-60: "As indicated by the name Domino, the N value from the N:M ratio decrements during the search phase with the weight penalty acting as momentum."
>
> **A10** Yeah we agree that this could cause some confusion. The idea of DominoSearch is that N will decrement during search like Domino game (e.g. N decrements as 10->9- >8->7 .. to represent the remaining rectangular tile in Domino game). We will think about a more straightforward name in revised submission. Suggestions are welcome.
>
> **Q11**: **Clarity:**  GDP and META are missing units in the FLOPs column in Table 2.
>
> **A11** Thanks for pointing out these issues. We will correct them in revised submission.
>
> **Q12**: **Significance:** extending to M=32 may not be reasonable from a hardware complexity standpoint. Reducing the size of M and showing compelling results would be a stronger argument for supporting such a design.
>
> **A12**: We agree that Extending M will introduce more hardware  overhead. We believe that it is important to explore the trade-off when designing a new hardware based on the design objectives (power, throughput and latency). The major overhead of extending M is that larger M will introduce more **index bits**. When M is 4, we need **2 bits** to encode one position (Assuming we use same encoding strategy in Nvidia Blogs[4]). When M goes to **16** we will need **4 bits** (2^4 = 16). If M goes to 32 then we will need **5 bits*** (2^5 = 32). Compared to the bit size of weight/data (float as 32-bit and half as 16 bit) and the memory footprint can be saved by not storing pruned weights, we think the extra storage for indices is affordable in many applications.
>
> Furthermore, we use M = 16 (4 bits, hardware prefers power-of-two numbers) in all experiments except for the two experiments (M = 32) in ResNet50 when going to sparsity = 93.75% because if we select M = 16 then we can only achieve this sparsity by setting all layers as 1:16 (sparsity of 1:16 is (1-1/16 = 93.75%)).
>
>
> [4] Ampere Sparse Tensor Core. https://developer.nvidia.com/blog/nvidia-ampere-architecture-in-depth/
>
> [5] Learning N:M Fine-grained Structured Sparse Neural Networks From Scratch, Aojun Zhou et al. ICLR2021
>
> [9] Rigging the Lottery: Making All Tickets Winners, Utku Evci et al. ICML2020
>
> [10] Soft Threshold Weight Reparameterization for Learnable Sparsity, Aditya Kusupati et al, ICML2020
>
> [14] Designing Network Design Spaces, Ilija Radosavovic et al. CVPR2020
>
> [15] Neural Pruning via Growing Regularization, Huan Wang, et al. ICLR 2021.
>
> [24] The State of Sparsity in Deep Neural Networks, Trevor Gale et al. ICML 2019
>
> *Hope our above responses are helpful to address your concerns. If you have further questions, please let us know. Thanks!*

---

> ### Author Response · Authors · 2021-08-10
> **Responses to Reviewer Mirr, Part 1**
>
>
> Thanks for your constructive comments to help us improve the quality of the work. We respond to your concerns as following:
>
> **Q1**: **Originality:** but I haven't seen a principled approach for N:M sparsity that doesn't rely on sweeping hyperparameters, etc. Similarly, pruning different layers to different amounts is nothing new, but it is when applied to N:M structured sparsity.
>
> **A1**: Indeed, we are the first to apply layer-wise approach to N:M sparsity . Regarding *principled approach* and *sweeping hyperparameters* we have three remarks:
>
> 1) Our method is a principled algorithm to address the problem of searching layer-wise N:M sparsity with predefined N:M candidates and complexity constraints(e.g. parameters, FLOPs, latency, power consumption) at affordable search cost (2-3 GPU hours). The effectiveness of working with parameters and FLOPs complexity has been demonstrated in the paper. Our method can be easily extended for other metrics such as latency and power consumption constraints with limited efforts (i.e. by look-up tables).
> 2) We do not sweep hyperparameters. We discuss the sensitivity of the important hyperparameter ($\beta$) introduced in **the search phase** in Appendix Sec3 and explain how it should be tuned for different complexity targets. For other hyperparameters such as **Kp**, we follow the setting in the corresponding literature[15].For training settings we adopt the hyperparameters in the SR-STE[5].
> 3) The existing layer-wise unstructured sparsity baseline -- STR[10], relies on more complicated hyperparameter tuning. The overall sparsity target is **implicitly** controlled by two hyperparameters **$\lambda$*** and **$s_{init}$** (Appendix A.6 in [10]). It means that for a new neural network and new sparsity target, the user first has to train multiple models with different $\lambda$ and $s_{init}$ to get desired sparsity/FLOPs target. ( e.g. **90% sparse ResNet50** : $\lambda$= 0.00003051757813, $s_{init}$=-800. **80% sparse ResNet50** : $\lambda$= 0.00001751757813 , $s_{init}$=-3200. **80% sparse MobileNetV1**: $\lambda$= 0.00001551757813 , $s_{init}$=-25). This hyperparameter tuning stage is time-consuming.
> By contrast, in our method, we can use the **same hyperparameter settings** for all network models and sparsity/FLOPs target.
>
>
>
>
> --------------------------------
>
>
> **Q2**: **Quality:** I'm concerned that the comparisons used to make claims like "… [the proposed result] is competitive to accuracy of the state-of-the-art layer-wise unstructured-sparsity model" aren't fair. Particularly, the problem being solved: DominoSearch is an algorithm to choose N:M values per layer of an already-trained dense network, but all comparisons are to algorithms which induce sparsity "from scratch," i.e. during the initial training process, without first training a dense network. Looking instead at a result for 80% sparsity (roughly matching the 25.6->5 Mparams result from Table 2) from Figure 5 in [1], the accuracy is 76.52%, barely reduced from the baseline of 76.69%. (This result is two years old, so the SOTA may have advanced since then.)
>
>
> **A2**: For the training settings we basically follow the settings of uniform N:M sparse training (SR-STE [5]), because it is our closest baseline. We agree that we should take the training time (e.g. fine-tune v.s. train-from-scratch) into consideration when comparing to other layer-wise unstructured sparsity especially STR[10]. Here we add additional experiments to show that even when we use fewer training epochs we can still achieve comparable results to STR[10]. We hope the additional experiments  can address reviewer's concern. Due to the limited time, we only conduct experiments for ResNet50 with 80% Sparsity.
>
>  | Index | Structure | Top1 |  Epochs| Train-from-scratch | Comments|
>  | ---- | -------- | ---------- | -----|-------- | -----|
>  | 1  | Layer-wise unstructured | 76.2      |     100  | Yes | Result of layer-wise unstructured sparsity [10]|
>  | 2  | Layer-wise N:M | 76.7      |     120  | No |  Result reported in submission|
>  | **3**|Layer-wise N:M |    76.4    |    120  | Yes |  Follow settings as SR-STE[5]|
>  | **4**|Layer-wise N:M |   76.1      | 100  | Yes |  Follow settings as STR[10]|
>  | **5**| Layer-wise N:M|  76.6  |    90  | NO |  Fewer Epochs|
>  | **6**|Layer-wise N:M |  75.8 |   90  | Yes | Fewer Epochs |
>
> * Result 1 (76.2) is the **layer-wise unstructured sparsity[10]** we compare to in our submission. Note that [10] has more non-standard hyperparameters (https://github.com/RAIVNLab/STR/blob/master/configs/largescale/resnet50-str.yaml).
> * Result 2 (76.7) is the one we report in our paper. The accuracy is 0.5 higher than result 1. *This is why we claim that our approach is able to achieve competitive results compared to the layer-wise unstructured sparsity*.
> * **Result 3** (76.4) is achieved by **training from scratch** with uniform N:M settings in [5]. The accuracy indeed decreases by 0.3 compared to result 2, it is competitive to result 1.
> * **Result 4** (76.1) is achieved by **training from scratch** with training settings in [10]. Although it is lower than result 1 by 0.1, it is till competitive to result 1.
> * **Result 5** (76.6) and **result 6** (75.8) use fewer epochs. Note that Result 6 (75.8) is much lower because **SR-STE[5] requires more epochs for convergence**. Result 5 indicates that we only need 90 epochs to reach the accuracy reported in result 2 which requires 120 epochs.
>
>
> The reason why we did not compare to the results of paper[1] or ref [24] in initial submission is that the experiments in [24] use very large batch size (**1024**). Note, experiments in [24] also use more  **1.5x** steps for sparse training.
>
> > While tuning the magnitude pruning ResNet-50 models, we observed that the best models always started and ended pruning during the third learning rate phase, before the second learning rate drop. To take advantage of this, we increase the number of training steps by **1.5x** by extending this learning rate region. -- section 5.2
>
> Besides, the layer-wise sparsity selection of paper [24] is  based on heuristics. The first layer has large contributions for FLOPs(due to large input feature map(224x224)) although it has relatively less contributions to #Params[10].
> > we modify our ResNet-50 training setup to leave the first convolutional layer fully dense, and only prune the final fully-connected layer to 80% sparsity. --section 5.2
>
>
> Therefore, when it comes to layer-wise sparsity comparisons, we compare our results to a more recent layer-wise unstructured sparsity work [10] which uses batch size of 256 (same as our setting). In order to make more fair comparisons, we will add the results of [24] and the table above in our revised submission.
>
> --------------------------------
> **Q3**: **Quality:** Does it make sense that accuracy for structured sparsity would be higher than accuracy for unstructured sparsity? If this is the case, then is it due to regularization, suggesting that the dense model was poorly trained to begin with? If, as you point out, unstructured sparsity is believed to be the upper bound, but you find that it isn't, I'd expect some discussion of why this commonly-held belief is not true.
>
> **A3**:  We did not claim that structured sparsity can always outperform unstructured sparsity and we indeed believe the accuracy of unstructured sparsity can be improved further. A key message of the comparisons is that the combination of proper structured N:M sparsity training algorithm(e.g. SR-STE[5]) and <u>proper layer-wise search algorithm (**our contribution**) is able to compete layer-wise unstructured sparsity</u>. This observation can **encourage** more hardware/system design efforts to speed up sparse network network.
> We agree that unstructured pruning should be theoretical upper-bound because it has **maximum** freedom for choosing which weights should be pruned/kept. However, this belief only holds when **1)** Better **pruning granularity** (e.g. larger M in N:M) always leads to higher accuracy or pruned model **2)** the **training algorithm** is able to train every weight sufficiently. Some previous works [e.g. [15]] have already shown that a sparse neural network is able to outperform the dense baseline(given the sparsity is small like 50% ). Lottery theory[9] also suggests that the current dense training algorithm may not be able achieve the full potential of dense network. In our paper, we did not contribute to the training algorithm, we follow the training algorithm of SR-STE[5]. **3)** The **layer-wise** sparsity scheme selection/search should be effective and can cooperate with the compression target, network architecture as well as training algorithm.
>
> We further explain these three points in the following.
> 1. **M** of N:M controls the pruning granularity or freedom. M = **16** could be sufficiently large (i.e. reach the roof-line of the benefit of pruning granularity) to achieve a competitive results compared to unstructured sparsity. For unstructured pruning, M can be interpreted as the number of parameters.
> 2. The training algorithm we adopt (SR-STE) is designed specifically for structured sparse training, so it may have some advantages when we compare to unstructured competitors [9,10] which generally adopt usual dense training setting (i.e. starts with dense training and induces sparsity gradually during training).
> 3. Our search method is able to find effective layer-wise sparsity within the pre-defined N:M candidates. This can be confirmed by comparing to uniform N:M baseline.

---

> > ### Comment · Reviewer_Mirr · 2021-08-17
> > **Still confusion about baselines**
> >
> > Hello, authors,
> >
> > Thank you for your responses.  I'm still not convinced that your comparisons to existing baselines are acceptable, though.
> >
> > ### SR-STE vs. ASP
> > You assert (in pat 2) that:
> > > Table 2 in [5] has demonstrated that SR-STE is a better training algorithm for N:M sparsity than Nvidia ASP
> >
> > You also said, in response to Reviewer WkEA, that:
> > > [1,5] have demonstrated that 50% sparsity will not cause accuracy drop
> >
> > Finally, consider two conclusions drawn from Tables 1 and 2 in [5], suggesting that the results shared for "ResNet50 ASP(Nvidia, 2020)" is not as expected:
> > 1. Accuracy is lost here for only 2:4 sparsity (76.8% vs. 77.3% top-1), which contradicts other published findings (and your own conclusions), and
> > 2. The epoch counts in table 2 suggest that the ASP result was not gathered with the suggested "repeat the dense baseline" again, or else the SR-STE results were allowed extra epochs.
> >
> > My conclusion is the opposite of yours: SR-STE has *not* been demonstrated to be a better training algorithm for N:M sparsity than that implemented in NVIDIA's ASP library.  I'm not arguing that it is better or worse, or that using SR-STE during fine-tuning will not help, just that the comparison presented is inconclusive without more details about the experimental procedure.
> >
> > ### Baselines
> > The real issue, then, is that the comparisons presented remain against training with sparsity "from scratch," rather than fine-tuning a dense model.
> > > For the training settings we basically follow the settings of uniform N:M sparse training (SR-STE [5]), because it is our closest baseline.
> >
> > Wouldn't your closest baseline be NVIDIA's ASP library, which uses a very similar methodology ("train dense, fine-tune")?  This seems to be much more similar to your approach than SR-STE, which is "from scratch."  The concern is not training time, but rather the number of training examples seen by the model during training; fine-tuning a dense model gets to see *many* more examples.
> >
> > Then, we come to this statement:
> > > Our search method is able to find effective layer-wise sparsity within the pre-defined N:M candidates. This can be confirmed by comparing to uniform N:M baseline.
> >
> > Unless the uniform N:M baselines underwent the same training schedule (train densely for X epochs, prune/determine masks/etc., fine-tune the mask for Y epochs), this conclusion does not hold.  As it is, I cannot tell how much better layer-wise N:M sparsity is compared to uniform N:M sparsity under the same training schedule.  Please let me know if I've missed such a comparison in your submission!
> >
> > ### SOTA
> > > The reason why we did not compare to the results of paper[1] or ref [24] in initial submission is that the experiments in [24] use very large batch size (1024). Note, experiments in [24] also use more 1.5x steps for sparse training.
> >
> > I agree that there were heroics used to train such an accurate model, but that is the nature of being state-of-the-art.  If you claim to be competitive with a state-of-the-art model (as you do in your abstract), then it is not right to compare to a model that merely closely matches your own methodology.
> >
> > Similarly, I agree that you never argued that structured sparsity can always outperform unstructured sparsity.  These are the sentences in the abstract that I take issue with:
> > > Furthermore, our layer-wise fine-grained N:M sparse ResNet50 achieves 76.7% top-1 accuracy with 5.0M parameters. This is competitive to 76.2% top-1 accuracy achieved by the state-of-the-art layer-wise unstructured-sparsity model with same number of parameters, which is believed to be the upper-bound of Neural Network pruning with respect to accuracy-sparsity trade-off.
> >
> > If you beat the SOTA by 0.5 percentage points, that'd be remarkable.  If the previous result were "believed to be the upper-bound," then I'd expect to see some discussion of this in the submission.  Your response suggests that you do not believe that unstructured sparsity is the upper bound?  In that case, it's strange to suggest the opposite in your abstract.

---

> > > ### Author Response · Authors · 2021-08-22
> > > **Responses to Reviewer Mirr - second round**
> > >
> > >
> > > Thanks for your constructive comments! We are happy that our previous responses can address some of your concerns.
> > >
> > >
> > >
> > >
> > > ## Q1 : SR-STE v.s. ASP
> > >
> > >
> > > We have no intention to argue which training algorithm is better or not. To verify that **layer-wise N: M (with the layer-wise schemes searched by our proposed method) is better than uniform**, we need to pick a training algorithm. SR-STE is the **most recent** training algorithm for N: M sparse training.
> > >
> > > More importantly, the **training algorithm** itself **is not** the key contribution of this submission. So we just pick a most recent one for experimenting layer-wise v.s. uniform.
> > >
> > > We agree that it is better to have additional experiments with two different training methods, which helps to reach a **stronger** and more **rigorous** conclusion.
> > >
> > > Following your suggestions, we conduct additional experiments with the ASP method, we will discuss these results later.
> > >
> > > ------------------------
> > >
> > > > You also said, in response to Reviewer WkEA, that:
> > > > [1,5] have demonstrated that 50% sparsity will not cause accuracy drop
> > >
> > > Apologize that we did not write it properly in the previous response. What we meant is that 50% sparsity will not cause a significant accuracy drop, so it is safe to start searching with 25% sparsity. We have corrected the sentence accordingly.
> > >
> > >
> > > ## Q2: Baselines
> > >
> > >
> > >
> > > In our first submission, we only perform experiments using SR-STE[5]. SR-STE is a relatively new (ICLR2021) method to train uniform N: M sparse neural networks. **We agree that SR-STE has not been extensively studied and verified because it is too new. We agree that using a well-studied and standard training method is better when comparing layer-wise v.s uniform.**
> > >
> > > Basically, there are two existing methods to train sparse N: M models：
> > > 1. **Fixed mask**: One-shot pruning and fine-tune, i.e. ASP[1].
> > > 2. **Dynamic mask**: Weights are allowed to regrow during fine-tuning/training, i.e. SR-STE[5].
> > >
> > > Following your suggestions, we conduct additional experiments using ASP to compare layer-wise v.s. uniform. Table 1 shows the results. Note that the experiments below have the same training setting as ASP:
> > >
> > > 1. Start with pre-trained models (from Pytorch model zoo).
> > > 2. Determine the mask based on pre-configured layer-wise/uniform sparsity.
> > > 3. Apply 100 epochs fine-tuning for accuracy recovery.
> > >
> > >
> > > *Note: the experiment schedules of both uniform and layer-wise are **exactly same** (from same pre-train dense model and fine-tune using ASP with 100 epochs)*.
> > >
> > > ### Table 1: Layer-wise v.s. Uniform using ASP
> > > | Index | Network | Method | Sparsity | Top-1 Acc (%) |  Epochs| Training Method |
> > > |-|:-:|:-:|:-:|:-:|:-:|:-:|
> > > | 1  |ResNet50 |Uniform 2:16 |87.5%   |   73.3   | 100 |  ASP|
> > > | 2  |ResNet50 |Layer-wise | 87.5%  |   74.9(**+1.6**)  | 100 | ASP |
> > > | 3  |ResNet18 |Uniform 2:16 |87.5%   |   64.2   | 100 |  ASP|
> > > | 4  |ResNet18 |Layer-wise | 87.5%  | 65.1(**+0.9**)    | 100 | ASP |
> > >
> > >
> > > The results of ASP **can support our claim** that our layer-wise N: M is significantly better than uniform N: M, which matches the observations under SR-STE training setting.
> > >
> > >
> > >
> > > We will add an ablation study section discussing the training methods in the revised version. We agree this discussion is very important.
> > >
> > > ## Q3: SOTA
> > >
> > > #### a) clarification
> > >
> > > We agree that the sentence - **76.2 achieved by STR is believed the upper bound** is an overclaiming and may cause misunderstanding. Specifically :
> > > 1. We agree that STR [10] is not the strict SOTA. Although it has closer experimental settings with ours, **76.5** achieved in [24] should be a stronger baseline. We will add this to our table in the revised version.
> > > 3. Unstructured sparsity is the theoretical upper bound, this is something we believe (We have more explanations in first-round response). We agree that the issue is that we did not compare to a stronger baseline (e.g. 76.5 in [24]) in our first submission.
> > >
> > > The corresponding sentences in the abstract are not rigorous and could cause misunderstanding. Reviewer-6z8t also pointed out this issue, and we replied that we will remove these sentences in the revised submission.
> > >
> > > #### b) Ablation study for the influence of training settings  and discussion of SOTA
> > > To study the influence of training settings and study how we get the high accuracy (76.7), we include different experiments for unstructured sparse ResNet50 and sparse N: M ResNet50 with the layer-wise sparsity searched by our algorithm in **Table 2**. The experiments in Table 2 consist of four parts:
> > >
> > >
> > > 1. Two layer-wise **unstructured sparsity** references [10,24]: **E1** and **E2**
> > > 2. The impact of the number of training/fine-tuning epochs: **E3 v.s. E4** and **E5 v.s. E6**
> > > 3. The impact of weight initializations (random v.s. pre-trained) : **E4 v.s. E5**
> > > 4. Different training methods (Dynamic mask(sr-ste[5]) v.s. fixed mask (ASP[1])): **E5 v.s. E7**
> > >
> > >
> > >
> > >
> > > ### Table 2: Influence of training settings
> > > | Index | Structure |Sparsity | Top-1 Acc (%) |  Epochs| Training Method | Weight initialization |
> > > |-|:-:|:-:|:-:|:-:|:-:|:-:|
> > > | E1  | Layer-wise unstructured |    80%   |    76.5   |  (~)150 (100x1.5) |[24]  |  Random |
> > > | E2  | Layer-wise unstructured |    80%   |    76.2   | 100 |STR[10]  | Random |
> > > | E3  | Layer-wise N:M  |  80%     |  76.1     | 100 |Dynamic mask[5]  |   Random |
> > > | E4  | Layer-wise N:M |  80%     |  76.4     | 120 |Dynamic mask[5]  |   Random |
> > > | E5  | Layer-wise N:M |  80%     |  76.7     | 120 | Dynamic mask[5] |pre-trained |
> > > | E6  | Layer-wise N:M |  80%     |  76.6     | 90 | Dynamic mask[5] |pre-trained |
> > > | E7 |Layer-wise N:M |   80%    |   75.6   | 120 | Fixed Mask[1] |   pre-trained |
> > >
> > >
> > >
> > >
> > >
> > >
> > > Discussions:
> > > 1. Layer-wise N: M v.s. Layer-wise unstructured:
> > >     * Compare E3(76.1) to E2(76.2): If using 100 epochs (same as STR[10]), our result is comparable to STR. Note that STR requires more complicated hyper-parameter searching as we explained in our first-round responses.
> > >     * Compare E4(76.4) to E2(76.2): If using 120 epochs (same as [5]) for training, layer-wise N: M can already outperform STR[10].
> > >     * Compare E4(76.4) to E1(76.5): We can achieve comparable accuracy to the stronger baseline using fewer epochs (120 v.s. ~150)
> > >
> > > 2. The impact of the number of training/fine-tuning epochs:
> > >     * Compare E4(76.4) to E3(76.1) and E5(76.7) to E6 (76.6): More training/fine-tuning epochs can help sparse networks to achieve better accuracy.
> > >
> > > 3. Weight initialization (random v.s. pre-trained):
> > >    * Compare  E5(76.7) to E4(76.4): Sparse training with pre-trained weights can lead to better results than that with random initialization.
> > >
> > > 4. Different training methods (Dynamic mask v.s. fixed mask):
> > >     * Compare E5(76.7) to E7(75.6): Layer-wise N: M sparse training with dynamic mask is better than with fixed mask.
> > >
> > > # Summary
> > > With the discussions and experiments above, we can conclude on how we achieve the SOTA result (76.7) for N: M sparse neural networks:
> > > 1. The layer-wise sparsity schemes searched by our algorithm is significantly better than the uniform counterpart. ( Verified under both ASP[1] and sr-ste[5], the **key contribution** of this submission)
> > > 2. We use pre-trained weights as initialization for magnitude pruning. [2]
> > > 3. We use sufficient epochs (120) for sparse training. [4]
> > > 4. We use the dynamic-mask-based training method to recover the accuracy [3,5]
> > >
> > >
> > > Finally, we would like to thank you again for your constructive suggestions on experiments. The experiments/ablation study are very useful and can improve our paper. We will add them to the revised submission.
> > >
> > >
> > >
> > > [1] Nvidia ASP, https://github.com/NVIDIA/apex/tree/master/apex/contrib/sparsity
> > >
> > > [2] Pruning Neural Networks at Initialization: Why Are We Missing the Mark? Jonathan Frankle et al. ICLR 2021
> > >
> > > [3] Dynamic Model Pruning with Feedback. Tao Lin  et al. ICLR2020
> > >
> > > [4] Rigging the Lottery: Making All Tickets Winners Utku Evci et al . ICML2020
> > >
> > > [5] Learning N: M Fine-grained Structured Sparse Neural Networks From Scratch, Aojun Zhou et al. ICLR2021
> > >
> > > [10] Soft Threshold Weight Reparameterization for Learnable Sparsity, Aditya Kusupati et al, ICML2020
> > >
> > > [24] The State of Sparsity in Deep Neural Networks, Trevor Gale et al. ICML 2019
> > >
> > > *Hope our above responses are helpful to address your concerns. If you have further questions, please let us know. Thanks!*

---

> > > > ### Comment · Reviewer_Mirr · 2021-08-24
> > > > **Thank you for recent experiments, please be specific when reporting results**
> > > >
> > > > Thank you, authors, for your continued explanations.  Your recent experiments with ASP have shown the results you initially set out to show, that, for some training schedule, an overall network sparsity of X% results in a higher-performing network using layer-wise N:M than with a uniform N:M.
> > > >
> > > > Please correct me if I'm wrong, but until these most recent experiments, there were no comparisons between layer-wise and uniform N:M with the same training schedule.  When comparing accuracy results, this difference is crucial.  As you showed in experiment (3), E5 vs. E4 shows that starting with a pre-trained network can have a material impact on final accuracy.  By picking SR-STE, which prunes from a random initialization during training, as a baseline, allowing the training schedule used to highlight your results to start with a *pre-trained* model gives your technique an unfair advantage.
> > > >
> > > > So, when presenting data to form a conclusion, it is imperative to only compare results that use the same training schedule (unless the schedule is the variable, as in experiment (3)).
> > > >
> > > > With the newest results, I increase my rating from (4) to (6).  Large changes will be necessary to make sure the comparisons reported in subsequent revisions are consistent as described above, though.

---

> > > > > ### Author Response · Authors · 2021-08-25
> > > > > **Thanks for your reviews and appreciation**
> > > > >
> > > > >
> > > > > Thank you for your appreciation to our new experiments and discussions. We are happy that they can help to address your concerns.
> > > > >
> > > > > We also would like to thank you for your suggestions on those new ablation experiments (strictly control the variables (e.g. epochs, training methods, pre-trained)). We recognize the importance of these comparisons.  We will include these discussions and ablation experiments in the revised submission.
> > > > >
> > > > > Regarding your question for comparisons between layer-wise and uniform N:M with the same training schedule:
> > > > >
> > > > > We included a table(**Table 1, in Q2 : Baselines**) in our **second-round responses**. The experiments compare Layer-wise v.s. Uniform under same training setting (ASP) following your previous suggestions.
> > > > >
> > > > >
> > > > >
> > > > > >### Table 1: Layer-wise v.s. Uniform using ASP
> > > > > >| Index | Network | Method | Sparsity | Top-1 Acc (%) |  Epochs| Training Method |
> > > > > >|-|:-:|:-:|:-:|:-:|:-:|:-:|
> > > > > >| 1  |ResNet50 |Uniform 2:16 |87.5%   |   73.3   | 100 |  ASP|
> > > > > >| 2  |ResNet50 |**Layer-wise** | 87.5%  |   74.9(**+1.6**)  | 100 | ASP |
> > > > > >| 3  |**ResNet18** |Uniform 2:16 |87.5%   |   64.2   | 100 |  ASP|
> > > > > >| 4  |**ResNet18** |**Layer-wise** | 87.5%  | 65.1(**+0.9**)    | 100 | ASP |
> > > > > >
> > > > > > **Note: the training schedules of both uniform and layer-wise are *exactly same* (from same pre-train dense model and fine-tune using the ASP method with 100 epochs)**, M is 16 in the experiments.
> > > > >
> > > > >
> > > > > We will organize these tables/experiments in a more well-structured and noticeable style in the revised version.
> > > > >
> > > > > **Please let us know if we misunderstood your question**.
> > > > >
> > > > >
> > > > > Thank you again for your constructive reviews and extensive discussions!
> > > > >
> > > > > Kind Regards,
> > > > >
> > > > > Paper4078 Authors.

---

### Official Review · Reviewer_WkEA · 2021-07-19

**Rating:** 6
**Confidence:** 5

**Summary:**

1) The paper proposes a novel layer-wise mixed N:M sparsification method, i.e., each group in a kernel can have different N. Compared to traditional layer-wise uniform sparsification methods, main techniques include group-wise threshold and layer-wise penalty (FLOPS and layer-wise redundancy).
2) Experiments on ResNet18 and ResNet50 show that for equal model size, the proposed methed achieves higher FLOPS compared to a most recent uniform N:M method [5]; for equal FLOPS, the proposed method achieves a smaller parameter number compared to [5].

**Limitations And Societal Impact:**

In Section 6, the authors state two limitations. For the first limitation on "did not benchmark deployment gains ...", I agree that adopting theoretical computation complexity (FLOPS) is an appropriate solution. For the second limitation about no experiments on Transformer, I think the current experiments on CNNs are sufficient already.

**Main Review:**

Pros.
1) The proposed method adopts a novel mixed N:M scheme, while a most recent N:M method [5] adopts uniform scheme. As shown in Experiments (Section 4.1), the proposed mixed scheme significantly improves compression ratio, compared to the traditional uniform scheme.
2) The proposed layer-wise penalty factor in Section 3.4 explicitly quantifies contribution of layer-wise complexity (FLOPS) and layer-wise redundancy, while previous methods [5,19] do not explicitly quantify the penalty factor as such.

Cons.
My major concern is that technical details of the proposed method are similar with recent N:M methods [5][19]:
1) In Section 3.2, the paper adopts "the mean of the smallest M/2 elements as threshold". However, a similar M/2 scheme is also adopted in a recent N:M method [19].  The authors need to explain differences between these two M/2 schemes.
2) In Section 3.3, the proposed weight update rule in Equation (4) is similar as the update rule adopted in a recent N:M method [5] (See Equation (4) in the paper [5]). Both these two update rules set regularization for the pruned parameters. The authors need to explain differences between these two weight update rules.

**Time Spent Reviewing:**

5

---

> ### Author Response · Authors · 2021-08-10
> **Responses to Reviewer WkEA**
>
> Thanks for your constructive comments to help us improve the quality of the work and your recognition to the pros of our submission. We respond to your concerns as following:
>
> **Q1**: The mean of the smallest M/2 elements as threshold". However, a similar M/2 scheme is also adopted in a recent N:M method [19]. The authors need to explain differences between these two M/2 schemes.
>
> **A1**: The purposes and values are different. We initialize the group-wise threshold with the mean of the smallest M/2 element, because it gives a good representation of group-wise threshold and also sufficient search spaces (see explanations in Line 134-144 ). For instance, if we have four values (0.0896 0.0169 0.0 0.0177, Figure 3 in our submission ), the threshold will be initialized as (0.0169+0.0)/2 = 0.00845. There are three (i.e. N = 3) weights will be kept based on this threshold, the search space is 3 out of 4 and starting sparsity is 1/4 = 25%. [1,5] have demonstrated that 50% sparsity will not cause significant accuracy drop, so it is safe to start searching with 25% sparsity.
>
> On the other hand. In [19], M/2 represents the number of weights should be kept because [19] focuses on 50% sparsity (i.e., N=M/2). In this case, the threshold is selected as the value of M/2(-th) smallest weight. In above example (0.0896 0.0169 0.0 0.0177), the threshold will be selected as second smallest weight(0.0169) which will give 50% sparsity (i.e., 0.0896 0.0177 will be kept ).
>
> --------------------------------
> **Q2**: the proposed weight update rule in Equation (4) is similar as the update rule adopted in a recent N:M method [5] (See Equation (4) in the paper [5]). Both these two update rules set regularization for the pruned parameters. The authors need to explain differences between these two weight update rules.
>
>
> **A2**:  In [5], the penalty is applied to weights which are supposed to be **pruned** given the fixed N:M. For instance, for 2:4 sparsity, weight penalty is applied to **smallest two** weights which are supposed to be pruned because of magnitude-based pruning criteria. The motivation of doing this in [5] is to stabilize and improve the sparse training. By contrast, Equation(4) in our submission applies layer-wise weight penalty to the **remaining weights**. The motivation is to introduce higher sparsity over the search stage to achieve the complexity target (e.g. #Params/FLOPs). Therefore, the formulations and purposes of these two equations are different.
>
>
> In summary. Although there are some similarities to related work[5,19], their purposes and technical details are totally different.
>
>
> *Hope our above responses are helpful to address your concerns. If you have further questions, please let us know. Thanks!*
>
> [1] Accelerating Sparse Deep Neural Networks. Asit Mishra et al. https://arxiv.org/pdf/2104.08378.pdf.
>
> [5] Learning N:M Fine-grained Structured Sparse Neural Networks From Scratch, Aojun Zhou et al. ICLR2021
>
> [19] Accelerated Sparse Neural Training: A Provable and Efficient Method to Find N:M Transposable Masks. Itay Hubara et al. https://arxiv.org/abs/2102.08124.

---

> ### Author Response · Authors · 2021-08-27
> **Would like to know your opinions on our explanations**
>
> Dear Reviewer WkEA:
>
>
> We thank you for your hard efforts spent on reviewing and your constructive comments. As the discussion period is approaching its end, we would like to know if our explanations have properly addressed your concerns on the differences between the notation (M/2) and equation(4) in this submission and related work [5,19].
>
> We always welcome your further comments and questions. We are always happy to provide our responses to them.
>
> Thank you.
>
> Kind Regards,
>
> Paper4078 Authors.
>
>
> [5] Learning N:M Fine-grained Structured Sparse Neural Networks From Scratch, Aojun Zhou et al. ICLR2021
>
> [19] Accelerated Sparse Neural Training: A Provable and Efficient Method to Find N:M Transposable Masks. Itay Hubara et al. https://arxiv.org/abs/2102.08124.

---

### Decision · Program_Chairs · 2021-09-27

**Decision:**

Accept (Poster)

**Comment:**

This paper provides a methodology for finding layer-wise fine-grained N:M sparse weights from a dense neural network. The reviewer discussion was very thorough, as was the authors' rebuttal. Overall, I think this paper is above the bar and I would like to see it published, although I think reviewer concerns about measuring against proper baselines should be addressed in the final version.